**Investigation**

# The distribution of beneficial mutational effects between two sister yeast species poorly explains natural outcomes of vineyard adaptation

Emery R. Longan (iD),[1,]* Justin C. Fay (iD)[1]

[1]Department of Biology, University of Rochester, Rochester, NY 14627, USA

*Corresponding author: Emery R. Longan, Department of Biology, University of Rochester, Rochester, NY 14627, USA. Email: elongan@ur.rochester.edu

Domesticated strains of *Saccharomyces cerevisiae* have adapted to resist copper and sulfite, 2 chemical stressors commonly used in winemaking. *S. paradoxus* has not adapted to these chemicals despite being consistently present in sympatry with *S. cerevisiae* in vineyards. This contrast could be driven by a number of factors including niche differences or differential access to resistance mutations between species. In this study, we used a comparative mutagenesis approach to test whether *S. paradoxus* is mutationally constrained with respect to acquiring greater copper and sulfite resistance. For both species, we assayed the rate, effect size, and pleiotropic costs of resistance mutations and sequenced a subset of 150 mutants. We found that the distributions of mutational effects displayed by the 2 species were similar and poorly explained the natural pattern. We also found that chromosome VIII aneuploidy and loss of function mutations in *PMA1* confer copper resistance in both species, whereas loss of function mutations in *REG1* was only a viable route to copper resistance in *S. cerevisiae*. We also observed a de novo duplication of the *CUP1* gene in *S. paradoxus* but not in *S. cerevisiae*. For sulfite, loss of function mutations in *RTS1* and *KSP1* confer resistance in both species, but mutations in *RTS1* have larger effects in *S. paradoxus*. Our results show that even when available mutations are largely similar, species can differ in the adaptive paths available to them. They also demonstrate that assays of the distribution of mutational effects may lack predictive insight concerning adaptive outcomes.

Keywords: *Saccharomyces cerevisiae*; *Saccharomyces paradoxus*; copper; sulfite; distribution of mutational effects; whole genome sequencing; phenotype assays

## Introduction

Adaptation is limited by the variation mutations can provide. Historically, it has often been asserted that mutations can yield variation along any dimension of an organism's phenotype (Lewontin 1974; Futuyma 1979; Futuyma 2010). This assumption in large part stems from the widespread success of artificial selection in laboratory environments and in domesticated plants and animals (Dobzhansky *et al.* 1977; Falconer and Mackay 1983; Hill and Caballero 1992). Consequently, influential models of adaptation such as Fisher's geometric model have focused on pleiotropy as the major source of evolutionary constraints rather than mutational availability (Fisher 1930; Orr 1998; Orr 2005). However, failures of the adaptive process due to lack of variation are not as uncommon as might be expected under this framework. For example, many plant species in England have failed to adapt to heavy metal contamination near copper mines whereas others have succeeded (Bradshaw 1984; Bradshaw 1991). Additionally, artificial selection experiments on some traits such as desiccation resistance in *Drosophila birchii* fail to elicit any response (Hoffmann *et al.* 2003). There are also intrinsic physical constraints on the structures of organisms that preclude improvement by mutation beyond certain thresholds (Alexander 1985). Failure of adaptation

due to lack of variation, termed "genostasis" or "sluggish evolution," challenges the idea that mutations can always enable adaptive evolution (Bradshaw 1991; Futuyma 2010).

The distribution of mutational effects (DME) is the distribution of effects for all possible mutations an organism or species can experience weighted by their frequency. In principle, the DME itself defines constraints on adaptation for the first step along an adaptive walk (Couce *et al.* 2024). Theory predicts that the distribution of effect sizes for beneficial mutations should be exponential (Gillespie 1991; Orr 2003; Eyre-Walker and Keightley 2007), and there have been several empirical studies which attempted to assay the DME, sometimes referred to as the distribution of fitness effects (Wloch *et al.* 2001; Sanjuán *et al.* 2004; Böndel *et al.* 2019; Couce *et al.* 2024). In general, what is found is that most mutations are deleterious or neutral, and a small fraction of mutations are beneficial to varying degrees. The small fraction of beneficial mutations is both critical to adaptation and very difficult to assay in most systems. This makes it hard to answer questions about the relative roles of chance vs adaptive potential when studying constraints in related species, particularly when one of the species adapts more successfully than another. As noted by Bradshaw (1991), "that what may occur in one organism, population, or species, may not occur in another," but the role of mutational availability in determining

these outcomes is only knowable if the beneficial tail of the DME is empirically measured.

Currently, there is a large gap in the understanding of how closely related species may differ in their adaptive capacity due to mutational constraints. This knowledge gap is particularly relevant to shifts in the environment that are universal to all resident species, such as changes in temperature or precipitation, or the introduction of a novel chemical stress. In such cases, there are many examples of successful parallel adaptation that span extreme phylogenetic distances, such as the evolution of cyclodiene resistance in four separate orders of insects by a parallel amino acid substitution (ffrench-Constant 1994). However, in cases like the metal-contaminated soils mentioned above, only a subset of plant species were able to successfully adapt to the excess copper introduced to the soil (Bradshaw 1984; Bradshaw 1991). As of now, evolutionary biology has very little power to predict if adaptation in any particular species will fail or succeed in these types of scenarios (Futuyma 2010). This is concerning given that the presence or absence of constraints can play key roles in many processes including extinction, pathogenicity, and viral immune evasion (Vogwill et al. 2014; Vogwill et al. 2016; Wu and Wilson 2017; Carabelli et al. 2023). Recent comparative experimental evolution studies have shown that related species often take similar, though not identical evolutionary paths toward adapting to a particular stress (Sanchez et al. 2017; Pentz and Lind 2021; Pentz et al. 2024). However, these studies are difficult to interpret from a mutational availability perspective because the outcome of experimental evolution is driven by interactions between mutation and selection (Van den Bergh et al. 2018).

In the present study, we used comparative mutagenesis as an alternative method to directly assay the mutational contribution to constraints in a pair of yeast species that differ in their apparent adaptive capacity to resist 2 anthropogenic stressors relevant to winemaking. Saccharomyces cerevisiae, unlike its sister species S. paradoxus, has successfully adapted to the enological stressors of copper and sulfite. These species are routinely found in sympatry in vineyard environments and oak forests (Sniegowski et al. 2002; Redžepović et al. 2002; Hyma and Fay 2013; Dashko et al. 2016; Vaudano et al. 2019). Domesticated strains of S. cerevisiae show much higher levels of copper and sulfite resistance compared to their wild counterparts (Pérez-Ortin et al. 2002; Fay et al. 2004; Warringer et al. 2011; Clowers et al. 2015; Dashko et al. 2016). S. paradoxus is widely considered a non-domesticated species, and despite being exposed to these chemicals in the vineyard environment, remains nearly universally sensitive to them (Liti et al. 2009; Warringer et al. 2011; Dashko et al. 2016; Yue et al. 2017).

Copper has been used as an agricultural antimicrobial since at least the mid-1700s (Money 2006; Borkow and Gabbay 2009) and came to prominence in viticulture in the 1880s when Pierre Marie Alexis Millardet published his experiments on the control of downy mildew and powdery mildew with a mixture of copper sulfate and calcium hydroxide which he called the "Bordeaux mixture" (Millardet 1885; Dixon 2004; Ayres 2004; Money 2006). Due to its low cost, high effectiveness, ease of use compared to other similar mixtures, and lack of adaptation in the problematic mildews it was intended to combat, this mixture has been sprayed in many vineyards yearly since the end of the 19th century (Masson 1887; Ayres 2004; Gessler et al. 2011). This has resulted in very high levels of copper in vineyard soils and alterations to microbial ecology in the affected areas (Besnard et al. 2001; Dell'Amico et al. 2008; Komárek et al. 2008; Mackie et al. 2012; Mackie et al. 2013; Fernández-Calviño and Bååth 2016; Grangeteau et al. 2017). The other primary use of copper in winemaking is the practice of adding copper sulfate to both red and white wines to remove sulfidic off-flavors, though this is not presumed to be a major source of selection on yeasts (Clark et al. 2015; Vela et al. 2017; Echave et al. 2021).

In the 1980s, it was discovered that high levels of copper tolerance in S. cerevisiae, a trait that is presumed to be uncommon in microbes (Borkow and Gabbay 2009), were due to tandem amplification of the CUP1 gene which encodes a short, cysteine-rich metallothionein that sequesters copper ions in the cytoplasm (Fogel and Welch 1982; Fogel et al. 1983; Capdevila et al. 2012). CUP1 copy number correlates quite strongly, though not perfectly, with copper resistance in S. cerevisiae (Fogel and Welch 1982; Fogel et al. 1983; Warringer et al. 2011; Adamo et al. 2012; Gerstein et al. 2015; Strope et al. 2015; Hull et al. 2017). Sequestration of excess copper is critical because of the detrimental effects free copper can have including disruption of iron-sulfur clusters (Macomber and Imlay 2009), destabilizing the cell membrane via lipid peroxidation (Avery et al. 1996), and damaging DNA (Tkeshelashvili et al. 1991). Interestingly, tandem amplification of this gene has independently occurred at least five times in S. cerevisiae and has never been observed in S. paradoxus (Welch et al. 1983; Zhao et al. 2014; Yue et al. 2017). Although CUP1 amplification is not the only genetic contributor to copper resistance in S. cerevisiae (Welch et al. 1989; Culotta et al. 1994; Gerstein et al. 2012; Chang et al. 2013), its absence in S. paradoxus likely explains a large fraction of the species difference.

Sulfite has been used as an antimicrobial in winemaking since at least the 19th century (Divol et al. 2012). Claude Ladrey made mention of burning sulfur in barrels as early as 1871 in his book about winemaking (Ladrey 1871; Divol et al. 2012). By the middle of the twentieth century, sulfite addition was common practice in the wine industry to suppress unwanted microbial growth during fermentation and prevent spoilage (Amerine et al. 1972; Jolly et al. 2006). There is evidence for the use of sulfur as an antimicrobial in winemaking long before this in ancient Egypt and in the Roman Empire, though the exact history is less clear than for copper (Pecci et al. 2020). Many domesticated strains of S. cerevisiae are very tolerant of sulfite due to the high expression of the SSU1 gene, which encodes a sulfite efflux pump. This high expression has evolved independently at least three times and, in all cases, involves structural variations that alter the upstream sequence of SSU1 (Goto-Yamamoto et al. 1998; Pérez-Ortin et al. 2002; Zimmer et al. 2014; García-Ríos et al. 2019). Similarly, S. uvarum, a more distantly related Saccharomyces yeast, has also convergently evolved greater SSU1 expression via rearrangements, but no known strain of S. paradoxus has done so (Macías et al. 2021). Of note, it has been shown recently that high expression of SSU1 has an intrinsic tradeoff with copper resistance due to Cup1 and Ssu1 having biochemically antagonistic roles in sulfur assimilation (Onetto et al. 2023). However, many domesticated strains retain resistance to both chemicals well beyond that of their wild counterparts (Dashko et al. 2016).

Sulfite tolerance is quite rare among other microbes (for some exceptions see Stratford et al. 1987; Varela et al. 2019). Although there is some variation within S. paradoxus, studies that compare the 2 species consistently find that S. cerevisiae is far more sulfite-resistant than the near-universally sensitive species S. paradoxus (Dashko et al. 2016). Given that S. paradoxus is routinely isolated from vineyards and wine must, these phenotypic differences suggest that S. paradoxus may be mutationally constrained with respect to its adaptation to sulfite.

Using this case of differential adaptation between microbial species as a model, we took a mutagenesis-based approach to

assay the DME between species to determine if differences in the DME may have predisposed the outcome seen in nature, namely *S. cerevisiae* repeatedly adapting and *S. paradoxus* repeatedly failing to adapt. If copper and sulfite resistance mutations are (1) more abundant, (2) of larger effect, or (3) tend to come with fewer pleiotropic costs in *S. cerevisiae* than in *S. paradoxus*, then the DME could explain why we see successful adaptation in the former species but not the latter. We used a mutagenesis-based approach rather than experimental evolution to circumvent the waiting time for adaptive variants to arise and fix, which is exacerbated in large asexual populations where clonal interference can greatly slow the rate of adaptation and fixation events (Gerrish and Lenski 1998; de Visser and Rozen 2006; Perfeito *et al.* 2007; Maddamsetti *et al.* 2015). Additionally, mutagenesis-based techniques allow for rare, large-effect mutations to be sampled more completely than in experimental evolution (Wloch *et al.* 2001).

In this study, we UV mutagenized copper- and sulfite-sensitive strains of *S. cerevisiae* and *S. paradoxus*. We then assayed mutational target size by measuring the frequency of resistance mutations when mutagenized cells were plated on various concentrations of these chemicals. We also subjected thousands of these mutants to a high-throughput phenotyping assay to measure the magnitude of resistance (mutational effect size) and any pleiotropic fitness costs in the absence of stress. A smaller subset ($N = 150$) had their genomes sequenced, and we identified causal variants underlying resistance in both species. Our results provide insight into species-specific mutations and effect sizes but highlight that the DME on its own may sometimes offer very little predictive or explanatory power in cases of apparent constraint.

## Materials and methods
### Yeast strains
We generated the focal strains used in this study from 2 *S. cerevisiae* and two *S. paradoxus* strains isolated from trees in Greece and France respectively (Robinson *et al.* 2016, Supplementary Table 1). The 2 *S. cerevisiae* strains are members of a Mediterranean oak population from which domesticated wine strains are thought to have been derived (Almeida *et al.* 2015). We knocked out the *HO* gene in monosporic derivatives of each of these four strains by transforming a natMX4 deletion cassette constructed using the plasmid pAG25 and primers that targeted the *HO* locus (Goldstein and McCusker 1999, Supplementary Table 2). We used primers JM7 and JM8 from Goldstein and McCusker (1999) for the construction of the *S. cerevisiae HO* deletion cassette. For *S. paradoxus*, we designed analogous primers with homology to the *S. paradoxus HO* locus (Supplementary Table 2). All transformations in this study were carried out using the lithium acetate/salmon sperm/PEG method (Gietz and Schiestl 2007). For all transformations into *S. paradoxus*, a modified heat shock of 40°C was used. We next sporulated transformants and dissected tetrads to obtain haploid derivatives of our ancestral strains. Successful *HO* integration was confirmed via co-segregation of nourseothricin resistance and haploidy assayed via PCR amplification of the MAT locus (Huxley *et al.* 1990, Supplementary Table 2).

For each monosporic clone, *MAT*a and *MAT*α haploid strains were obtained (Supplementary Table 1). These haploids were then mated to their isogenic counterparts on YPD (Bacto yeast extract 10 g/L, Bacto peptone 20 g/L, dextrose 20 g/L, and agar 20 g/L) plates to obtain homozygous diploids, which were confirmed as diploid via PCR amplification of the MAT locus and the capacity to sporulate. Following these manipulations, our focal strain set for this study included 12 strains derived from the 4 wild isolates:

A *MAT*a haploid derivative, a *MAT*α haploid derivative, and a homozygous diploid derivative of each (Supplementary Table 1). In our phenotyping assays, derivatives of the laboratory *S. cerevisiae* strain S288c (YJF5538) and a North American oak *S. cerevisiae* strain, YPS163 (YJF5539), were included as points of reference (Mortimer and Johnston 1986; Sniegowski *et al.* 2002).

In addition to the focal strains, 150 mutants derived from these strains were sequenced and are listed in Supplementary Table 3. For follow-up assays on *PMA1* mutations, 5 heterozygous diploid strains for each species were constructed by mating sequenced mutants to their *MAT*α parent (Supplementary Table 1). For follow-up assays on *REG1*, deletion strains of both species were constructed via transformation of an hphMX4 cassette targeted to the *REG1* locus (Supplementary Table 2). Deletion mutants were confirmed via PCR. A suppressor mutant of the slow growth phenotype seen in the *S. paradoxus REG1* deletion strain was also recovered and included in the *REG1* follow-up assays (Supplementary Table 1). Information concerning the other mutants used in the high-throughput phenotyping and the data relevant to them can be found with the following DOI: 10.6084/m9.figshare.25777512.

### Mutagenesis
We mutagenized two haploid (*MAT*a) and 2 diploid strains from each species. Our mutagenesis protocol is similar to the irradiation protocol found in Birrell *et al.* (2001). We used 2 mL of saturated overnight culture (YPD) as our input into mutagenesis and our control mock mutagenesis for each strain. This culture was spun down at 700 rcf for 5 minutes and was resuspended in 10 mL of autoclaved deionized water. This suspension was then transferred to a petri dish and subjected to either 0 seconds (mock mutagenesis) or 10 seconds of UV-C radiation in a Singer Rotor HDA (Singer Instruments, Somerset, England). The cell suspension was then moved back to a 15-mL conical tube and spun down at 700 rcf for 5 minutes. Next, we resuspended the mutagenized and control pools into 6 mL of YPD and allowed the cultures to recover overnight at 30°C with shaking at 250 rpm and then stored these cells the following day as 15% glycerol stocks at −80°C.

### Mutant isolation
To isolate copper and sulfite mutants, we plated the mutagenized and mock mutagenized pools onto solid media containing added copper or sulfite. Specifically, 200 µL of the mutagenized and control pools for both the haploid and diploid strains was plated on eight concentrations of complete media (CM: US Biological drop-out mix complete without yeast nitrogen base 1.3 g/L, Difco yeast nitrogen base 1.7 g/L, ammonium sulfate 5 g/L, dextrose 20 g/L, and agar 20 g/L) + copper sulfate (0.0 mM, 0.05 mM, 0.1 mM, 0.15 mM, 0.2 mM, 0.3 mM, 0.4 mM, 0.5 mM, and 0.6 mM). Additionally, we plated the pools on eight concentrations of CM + 75 mM tartaric acid/sodium tartrate (pH = 3.5) with various amounts of added 0.5 M sodium sulfite. The final concentrations were 0 mM, 0.5 mM, 1 mM, 1.5 mM, 2 mM, 3 mM, 4 mM, 5 mM, and 6 mM sulfite. Notably, sodium sulfite has long been known to lack thermostability due to the off-gassing of the main cytotoxic sulfite species, $SO_2$, complicating the formation of solid sodium sulfite plates. Our method was to add sodium sulfite to 50-mL aliquots of hot autoclaved agar media and pour the plates immediately. This consistently led to the toxicity of sulfite being retained and to the apparent uniformity of sulfite concentrations within single plates as compared to the potentially noisier methods that have been used in the past such as spreading sodium sulfite onto solidified plates (Park *et al.* 1999). To control for cell

density, 200 μL of a $10^{-5}$ dilution of each mutagenized and mock mutagenized haploid pool was plated onto YPD. Plating 200 μL of a $10^{-4}$ dilution was used for this purpose for diploids. After plating, plates were incubated at 30°C for five days and the number of colonies per plate was counted (Supplementary Table 4).

## Mutation rate calculation

We estimated the induced mutation rate in our haploid mutagenized pools by plating mutagenized and control pools of each haploid strain on three concentrations of CM + canavanine (30, 45, and 60 mg/L). The mutant induction rate was calculated as the number of colonies recovered divided by the total number of cells plated for each strain on each concentration of canavanine. To assay differences among strains, we compared our canavanine mutation rates across the 3 concentrations between species using a $t$-test with mutation rates paired by concentration. We also compared mutation rates at the highest concentration with a chi-square test. Next, using canavanine mutation rate estimates from Lang and Murray (2008), we estimated the mutations per genome in our mutagenized pool along with the saturation of our screen (see also Gruber *et al.* 2012; Metzger *et al.* 2016, and Hodgins-Davis *et al.* 2019 for similar calculations). Using a canavanine plating assay; Lang and Murray (2008) estimated the number of spontaneous mutations per base pair per cell division as $6.44 \times 10^{-10}$ based on the rate at which they recovered mutants with a canavanine resistance phenotype, which was $1.52 \times 10^{-7}$. We used these rates to estimate the number of induced mutations per genome by multiplying this per base pair mutation rate by the fold increase in phenotypic mutation rate we observed in our mutagenized pools on 60 mg/ml canavanine relative to Lang and Murray's phenotypic mutation rate (Supplementary Table 4). Across the four strains, we saw an average of an 88-fold increase in phenotypic mutation rates yielding an estimate of $5.66 \times 10^{-8}$ mutations per base pair. When multiplied by the length of the genome this yields an average of 0.68 mutations per genome in our mutagenized pools. When our inferred per base pair mutation rate is multiplied by the number of cells plated in each screen it yields an average of 1.7 mutations per base pair in the genome for copper and 1.22 mutations per base pair in the genome for sulfite. These numbers refer to the average number of mutants in the whole screen at any given site in the yeast genome (Supplementary Table 4). Critically, these numbers do not take into consideration the biased mutation spectrum of UV mutagenesis (Mao *et al.* 2017). For our plating assays on copper and sulfite, mutation rates were simply calculated as the number of colonies recovered divided by the total number of cells plated for each strain on each concentration. These rates were compared from species to species using pairwise chi-square tests at each plating concentration among the four ancestral strains for copper. All plating concentrations were collapsed to a single chi-square test for sulfite given the much lower number of mutants recovered. Colony counts and inferred saturation calculations can be found in Supplementary Table 4.

## Mutant picking

A subset of haploid ($N = 3,024$) and diploid ($N = 720$) mutants that produced colonies on stress concentrations that killed the ancestor were sampled for follow-up phenotyping to assess the effect size and pleiotropic consequences of the mutations. Sampling was done by recovering single colonies via pipette tips and manually arraying cells from these colonies in a grid on a solid CM plate. For haploid copper mutants, we sampled 1,512 resistant isolates and arrayed them in 4 (one for each ancestral strain) 384 format

plates along with 6 controls per plate. These controls were YJF5538 (S288c derivative), YJF5539 (YPS163 derivative), 2 replicates of the ancestral strain, and 2 independent isolates recovered from the mutagenized pool without selection to control for the effect of mutagenesis. The mutants consisted of 168 spontaneous mutants and 1,344 induced mutants. For haploid sulfite mutants, the same procedure and controls were used to array 1,025 spontaneous mutants and 487 induced mutants. Regarding sulfite mutants, both *S. paradoxus* strains produced fewer than 378 mutants ($N = 330$ for YJF3734 and $N = 110$ for YJF3815). Thus, the remaining positions on these plates were filled with *S. cerevisiae*-derived sulfite mutants. After initial arraying on 384 plates, these were collapsed to 1536 format solid plates to await phenotyping. Supplementary Fig. S1a summarizes the mutant picking pipeline.

For diploid copper and sulfite mutants, we similarly sampled 90 resistant isolates for each of the strains from the plating assay. These were arrayed in 96 format along with six controls, which were four replicates of the ancestor and a replicate each of YJF5538 and YJF5539. For both copper and sulfite, these mutants comprised 48 spontaneous mutants and 312 induced mutants split evenly among the four ancestral strains. These plates were collapsed to 1536 format with 2 technical replicates of each mutant on the plate to await phenotyping. For both diploids and haploids, mutants appearing on higher concentrations of copper and sulfite were prioritized and sampled exhaustively because these were putatively the largest effect mutants. We sampled as evenly as our plates permitted across the lower concentrations. For each mutant, we recorded recovery concentration and three aspects of colony morphology when picking them from the stressor plate: relative size, color, and circularity. These data can be found at DOI: 10.6084/m9.figshare.25577512 . In total, 1,872 mutants were selected for phenotyping for each stressor, and this consisted of 1,512 haploid mutants and 360 diploid mutants.

## Phenotyping

Arrayed strains were phenotyped on copper, sulfite, and six permissive conditions. For both haploid copper and sulfite mutants, a master 1536 plate (CM) that was grown for 2 days harboring all of the mutants and controls was replica plated onto phenotyping plates. Replica plating was performed using a Singer Rotor HDA robot (Singer Instruments). For copper, we used 18 concentrations ranging from 0 mM to 0.8 mM, and for sulfite, we used 16 concentrations ranging from 0 to 6 mM for haploids and 21 concentrations ranging from 0 to 6 mM for diploids. Our permissive conditions encompassed three different base media: YP, CM, and minimal media (MM: yeast nitrogen base 1.7 g/L, ammonium sulfate 5 g/L) with either dextrose (2%) or glycerol (3%) as a carbon source. All stressor and permissive conditions were incubated at 30°C.

Colony size was measured using images obtained in a Singer Phenobooth using the Phenosuite software package (Singer Instruments). Colony size measurements were performed at 24 hour intervals for 3 days for copper and permissive conditions. For sulfite, we imaged the plates for five days because a growth delay is a common phenotype associated with sulfite stress. Colony sizes are recorded as the number of pixels the colony occupies in an image. Example raw and processed images are shown in Supplementary Fig. 2. For haploids, images were taken at 1280 × 960 resolution, and for diploids, 4128 × 3096 resolution was used.

After imaging, we obtained a total of 522,240 colony size measurements in the haploid dataset and 299,520 in the diploid dataset. Colony size measurements were subjected to manual quality control, and technical replicates in the diploid dataset were

averaged to a single measurement. During manual curation, we removed 63,382 and 9,784 colony size measurements from the haploid and diploid datasets respectively. These removals were due to failure of a colony being transferred to a plate or excessive drying of an area of a plate. For sequenced mutants, copper and sulfite phenotypes were confirmed via repeating the phenotyping protocol in triplicate in 384 format. All three phenotyping datasets for the haploid, diploid, and sequenced mutants can be found at DOI: 10.6084/m9.figshare.25777512.

## Colony size analysis

For copper and sulfite, we measured resistance by the area under the curve (AUC) as a function of concentration. AUC was estimated using the trapezoid rule implemented in the R package Pracma (Borchers 2019). Each colony size measurement on a stressor plate was normalized to the relevant no-stress condition for each mutant. The AUC phenotyping method is summarized in Supplementary Fig. S1b. Comparisons in the AUC metric were made using ΔAUC relative to the ancestral AUC via Kruskal–Wallis tests. For our triplicate assays on the sequenced mutants, we used the same method and averaged the technical replicates. Permissive phenotypes were quantified as growth relative to the ancestor. Thus, a score of 1 means that a strain produced a colony of equivalent size as compared to its ancestor. Edge effects (Baryshnikova et al. 2010; Zackrisson et al. 2016) were removed from the permissive growth data by a "layer" based normalization (Miller et al. 2022). This normalization entails grouping colonies by the number of positions that separate them from the edge of the plate. On a 1536 plate, this yields 16 distinct layers. Row and column normalization was not possible because row is confounded with species in these experiments.

Within our copper and sulfite data, several "mutants" appeared to phenocopy their ancestor in stress conditions. This indicated that these strains produced colonies in our plating assay for reasons other than harboring a mutation conferring heritable stress resistance. We categorized these strains as "physiological escapees" and defined them statistically as any mutant failing to yield an AUC not exceeding three standard deviations greater than that of the ancestral mean. These "escapees" were removed from all analyses of our assays. For haploid sulfite mutants, we were only able to rule out our sequenced mutants as non-escapees (see below), and these strains' triplicate phenotypes were the only ones retained in those colony size analyses.

## Genome sequencing

For copper mutants, we selected 120 mutants for sequencing, evenly distributed across our 4 haploid ancestral strains. We focused genomic analyses on haploids to avoid confounding effects of dominance and to facilitate the identification of mutations (e.g. eliminating many false positive heterozygous calls). For mutants derived from each ancestor, we divided mutants into quartiles based on their copper resistance (ΔAUC). The bottom quartile was removed from consideration, and 10 mutants were randomly selected from the top 3 quartiles for each ancestral strain's haploid derivatives. For sulfite mutants, the top 30 AUCs for each haploid ancestral strain's derivatives were chosen as sequencing candidates. After triplicate phenotyping, 31 strains remained as non-escapees, 12 from S. cerevisiae and 19 from S. paradoxus. Both spontaneous and induced mutants from a wide variety of recovery concentrations were sequenced (Supplementary Table 3), but selection for sequencing was done independent of these variables and only considering ΔAUC.

Following selection, genome sequencing was completed for these 151 haploid mutant strains and each of the four ancestral strains. One copper mutant had very low coverage and was eliminated from further analysis. For each of the strains that were subjected to whole genome sequencing, we extracted genomic DNA using a Yeastar genomic DNA kit (Zymo Research), prepared our libraries using a Nextera DNA Flex Library Preparation Kit (Illumina), and sequenced them as multiplexed libraries on a Hiseq X platform (150 bp, paired-end) via Novogene.

We mapped reads for each strain to the S. cerevisiae or S. paradoxus reference genomes (S288C_reference_sequence_R64-2-1 and S. paradoxus ultrascaffolds retrieved from the Saccharomyces sensu stricto database) using version 0.7.17 of the Burrows-Wheeler aligner (Scannell et al. 2011; Cherry et al. 2012; Li 2013). Reads were marked duplicate with Picard tools version 2.12.0, and variants were called using version 4.1.7.0 of GATK using the HaplotypeCaller command. We applied six filters to our dataset: (1) Sites that had 4 or more unique genotype calls across all sequenced strains were removed due to low confidence. (2) Because all sequenced strains are haploid, sites lacking 2 different homozygous genotype calls were removed. (3) Sites where the ancestral strain had a heterozygous call were removed. (4) If the ancestral genotype call disagreed with all other calls at the site, that site was removed. (5) If > 10% of the derivatives of any ancestor lack a call at a site, that site was removed only for that ancestral strain and its derivatives. (6) If a single site had more than one insertion/deletion (indel) called in multiple strains or had a genotype quality less than 10, it was removed due to low confidence. After applying these filters for the S. cerevisiae mutants, 700 sites were retained across the derivatives of both ancestral strains out of 91,840 SNPs and indels. After applying these filters for the S. paradoxus mutants, 334 sites were retained across the derivatives of both ancestral strains out of 33,723 SNPs and indels. Details can be found in Supplementary Table 5.

Mutations were annotated for their functional effects on protein sequence using SNPeff (Version 4.3t, build 2017-11-24 10:18) along with genome annotation files downloaded from the Saccahromyces Genome Database (S288C_reference_sequence_R64-2-1) and the Saccharomyces Sensu Stricto Database (Scannell et al. 2011; Cherry et al. 2012). Of note, PMA1 and PMA2 are erroneously swapped in the annotations in the Saccharomyces sensu stricto database for S. paradoxus. We corrected this issue for our identification of mutants.

Structural variation was assessed using the program Delly (version 0.8.7). We applied a similar set of filters to our structural variant calls as our SNPs and indels. Additional filters for low-quality and imprecise calls were added, and we altered the requirement for 2 different homozygous calls to exclude duplications, as these may be expected to be called heterozygous. To be specific, this alteration involved removing any calls that were not singletons and removing any heterozygous singletons that were not duplications. We also manually inspected the remaining variants for anomalous patterns such as the majority of strains being called heterozygous for a variant. Of a total of 51,804 variants, only 5 were retained after these filters were applied (Supplementary Table 5). To ensure that CUP1 tandem duplications were not missed, we manually inspected coverage data at the CUP1 locus for all copper mutants. For the strain YJF4464, which harbors a CUP1 duplication, we also mapped reads to a modified reference sequence containing an inverted repeat of the CUP1 region to determine the structure of the repeat. The SSU1 locus was also manually inspected in all sequenced sulfite mutants, and the FZF1 locus was analyzed and inspected accounting for the introgressed S. paradoxus allele present in YJF3731 and its derivatives. After PMA1 was

identified as causal for copper, manual inspection of this gene yielded the discovery of one additional mutation that was previously missed in YJF4433 and 2 synonymous mutations that were missed in YJF4439. These post hoc findings are noted in Supplementary Table 3.

Karyotypes for each strain were obtained systematically via sequencing depth. Specifically, for each strain, depth was calculated for each 1-kb window across the genome, and these numbers were averaged for each chromosome. Then, the depth for the individual chromosomes was normalized to the coverage of the least-covered chromosome. Relative read depth was then assessed and an output for the chromosome count for each of the 16 chromosomes was obtained assuming there is a single copy of the least-covered chromosome. Relative read depth was calculated disregarding the rDNA cluster on chromosome XII. These karyotypes were then manually inspected and confirmed via assessment of the genome-wide coverage. To account for the possibility of diploidization (Mable and Otto 2001; Zeyl *et al.* 2003; Gerstein *et al.* 2006; Selmecki *et al.* 2015), separate estimates of chromosome number were obtained assuming a copy number of 2 for the least-covered chromosome. A strain was considered as having evidence for diploidization if more than one aneuploid chromosome showed coverage values consistent with the diploid estimates. Nine strains had a coverage pattern consistent with diploidization and this is noted in the karyotype calls (Supplementary Table 3). Three strains had anomalous coverage patterns not easily accounted for by diploidization, and these are also noted in Supplementary Table 3. Manual inspection did not reveal any cases of partial chromosome loss and revealed only one clear case of partial chromosome gain in S. *paradoxus* strain YJF4480 (Supplementary Table 3).

## Simulations to assess gene/aneuploidy significance

To determine which genes were significant hits in our mutant screen, we performed *in silico* simulations of our experiment using the number of mutations detected in our dataset that alter protein sequences. Specifically, we used our empirical number of disruptive mutations (nonsynonymous, frameshift, nonsense, in-frame indels, and stop-lost mutations) as input for these simulations along with the length of every gene in the genome for each species. The simulation "rains down" a number of mutations equal to the number observed in our experiments onto the coding sequences of the genome and counts the number of hits found in each gene. By performing this simulation 1,000,000 times per species, we arrived at a null distribution for the number of mutations expected for each gene accounting for gene length. This in turn yields an empirical *P*-value for each gene as the fraction of simulations with equal or greater hits than the number of hits in the dataset. We assessed the significance of each gene by applying a Bonferroni correction ($N = 6,696$ for S. *cerevisiae* and $N = 5,963$ for S. *paradoxus*) to these empirical *P*-values. For both species, information for each gene with multiple hits in our experiment, and its significance can be found in Supplementary Table 6.

A similar procedure was carried out for chromosome aneuploidies. We used the number of aneuploid strains and the number of aneuploid chromosomes in each of these strains as input. For each iteration of the simulation, this number of strains with the specified number of aneuploid chromosomes had chromosomes randomly selected as aneuploid. To illustrate, for S. *cerevisiae* copper mutants there were eight total aneuploid strains. Four of these strains had one aneuploid chromosome, and the other four had 2,

3, and 7 aneuploid chromosomes, respectively. For each iteration of the simulation, 8 strains with these numbers of aneuploidies had their aneuploid chromosomes selected at random, and these simulated data were used to give an empirical *P*-value for each chromosome. To test for enrichment of specific chromosome combinations, namely Chromosome III and Chromosome VIII, we simulated the number of aneuploidies observed in our dataset for each chromosome and counted the number of strains harboring both aneuploidies for each iteration. These empirical *P*-values were used to assess significance. Incidence data concerning all significant genes and aneuploidies are summarized in Supplementary Fig. 3. Statistics comparing incidence, effect sizes, and costs between species for significant genes and aneuploidies are summarized in Supplementary Table 7.

## PMA1 pH sensitivity and dominance assay

To test the effects of *PMA1* mutations on low pH sensitivity, we spot-diluted 39 *PMA1* mutants onto MM and low pH MM (MM + 10 mg/mL unbuffered tartaric acid, pH = 2.5). Cells from an overnight culture were serially diluted on each medium. Mutants were qualitatively scored based on their relative growth on the 2 media relative to the ancestor as either low pH sensitive or not low pH sensitive (Supplementary Table 8). Another follow-up was carried out to determine if *PMA1* mutations are dominant or recessive with respect to their effect on copper resistance and acid sensitivity. For five of the six cases where S. *cerevisiae* and S. *paradoxus* yielded site-level parallel changes in *PMA1*, diploid heterozygotes (Supplementary Table 1) were constructed and spot-diluted as described above onto MM, low pH MM, CM, and several concentrations of CM + CuSO$_4$ (0.02 mM, 0.05 mM, 0.1 mM, and 0.2 mM). The heterozygotes were scored qualitatively relative to their ancestor and their haploid counterparts (Supplementary Table 9). An additional summary of all the amino acid changes predicted from recovered mutations in *PMA1* along with their known phenotypes (Young *et al.* 2023) in prior studies can be found in Supplementary Table 10.

## REG1 deletion and spot dilutions

Following sequencing, *REG1* was chosen for follow-up experiments using the deletion mutants described above. To test for the phenotypic effects of *REG1* deletion in S. *cerevisiae* and S. *paradoxus*, we performed spot dilution assays on four separate media for constructed deletion strains and their ancestors. The media used were CM, CM + 0.1 mM CuSO$_4$, MM, and low pH MM. Cells were spot-diluted on each medium as described above and imaged using a Phenobooth following 3 days of growth (Singer Instruments). YJF5535 (S. *paradoxus* Δ*reg1*) also had a 10× concentrate of its overnight culture included on the plates due to its poor growth rate. Additionally, several sequenced *REG1* mutants (YJF4444, YJF4445, and YJF4452) were also assayed for their sensitivity to low pH media as described above.

## Results

To test whether differences in adaptation to vineyard stressors between S. *cerevisiae* and S. *paradoxus* can be explained by differences in their DME, we UV mutagenized haploid and diploid derivatives of 2 Mediterranean oak strains of both species and isolated mutants with elevated copper and sulfite resistance. If resistance mutations in S. *paradoxus* have a smaller mutational target size, are of smaller average effect, or come with greater costs in permissive conditions when compared to S. *cerevisiae*, the DME may explain the apparent constraints seen in nature.

## Mutant screen

To test the mutational target size for resistance mutations in both species, we screened our mutagenized pools by plating them onto several concentrations of copper and sulfite. We also plated our haploid mutagenized pools onto three concentrations of canavanine to calculate an induced mutation rate and the per base pair genome-wide saturation of our screen (see Methods). In our canavanine control, we find that spontaneous and induced phenotypic mutation rates did not differ between species across our three assayed concentrations (Supplementary Fig. 4, paired t-test, $P > 0.50$ in both cases) or when only the highest canavanine concentration is considered (chi-square test, $P = 0.85$). This is consistent with equal induction of mutations between the 2 species. We used data from our highest canavanine concentration to estimate the genome-wide saturation of our mutant screen (see Methods). These data and calculations (Supplementary Table 4) give rise to an expectation that each gene was mutated many times and site-level parallelism had a reasonable probability of occurring.

## Mutational target size

To estimate rates of copper and sulfite resistance mutations, we plated mutagenized and mock mutagenized cells on plates with a range of copper and sulfite concentrations. There were some differences between species at low copper concentrations in the recovery rates of mutants, but no differences were observed for sulfite (Fig. 1). On copper, we obtained 8,704 mutants, mostly from the lowest concentrations. For several of the lower concentrations, both S. cerevisiae strains produced mutants at a higher rate than both S. paradoxus strains did (chi-square tests, Bonferroni corrected, $P < 0.05$ in all cases noted in Fig. 1a). At the higher concentrations, this pattern was not observed.

Importantly, the measurements at low concentrations are potentially susceptible to bias driven by slight differences in basal resistance in the ancestors and are more prone to yielding false positive colonies ("physiological escapees," see Methods and below). As such, these results support similar target sizes between species, but point to the potential of a target size difference favoring S. cerevisiae specifically at low copper concentrations.

At low sulfite concentrations, we observed far more S. cerevisiae colonies than S. paradoxus colonies (Supplementary Table 4). However, unlike for copper, upon phenotyping we found that the vast majority of these "mutants" did not have a heritable sulfite resistance phenotype (see Methods). We defined these isolates as "physiological escapees." After retroactively removing escapees, we retained only 31 haploid and 7 diploid mutants, and when considering only these isolates there were no differences in rates of induced or spontaneous mutants between species for either haploids or diploids (Fig. 1b, chi-square tests, $P > 0.15$ in all cases). Taken together, the results from our initial mutant screen are not consistent with large differences in mutational target size.

## Mutant effect sizes

To test whether mutational effect size among mutations conferring elevated resistance to copper and sulfite differs between S. cerevisiae and S. paradoxus, we subjected a large subset of the mutants identified in our screen to a high-throughput, robotics-based phenotyping assay. Briefly, we arrayed our mutants ($N = 3,024$ for haploids and $N = 720$ for diploids split evenly between copper and sulfite) onto solid plates containing many different concentrations of copper and sulfite. Our subset contained mutants from both the mutagenized and mock mutagenized pools isolated at various concentrations of copper and sulfite. Specifically, we

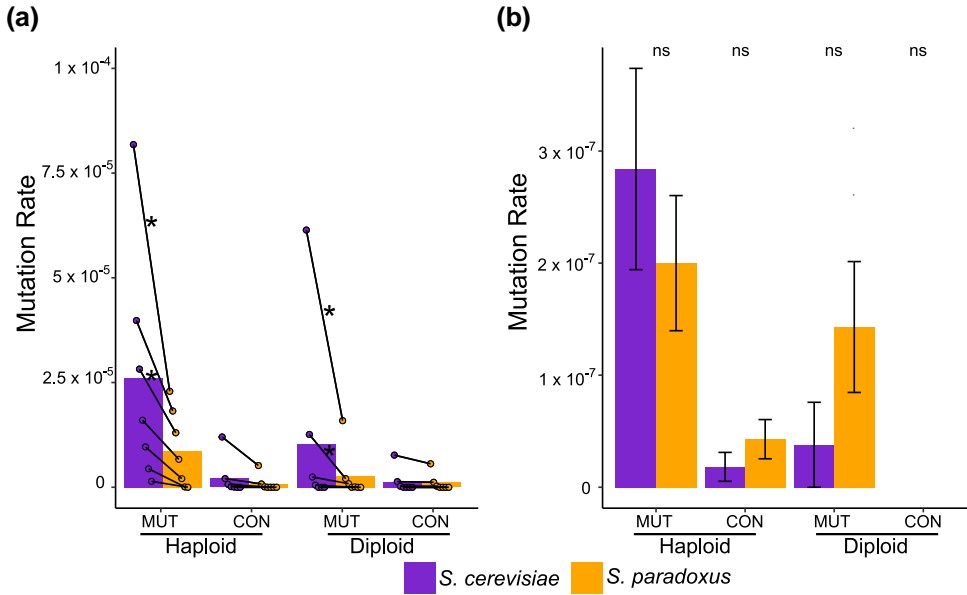

**Fig. 1.** Mutation rates to copper or sulfite resistance between species. Mutation rates in platings of mutagenized and mock mutagenized pools of S. cerevisiae and S. paradoxus. a) Copper mutation rates for haploids and diploids as measured by the number of colonies divided by the number of cells plated. Points represent individual measurements for different copper concentrations and are paired by concentration. Left to right, points signify mean mutation rate for the two ancestor strains on 0.1 mM, 0.15 mM, 0.2 mM, 0.3 mM, 0.4 mM, 0.5 mM, and 0.6 mM $CuSO_4$. Bars signify overall mutation rate with all concentrations pooled together. An asterisk denotes a significant difference in mutation rates for a concentration such that both S. cerevisiae strains produced mutants at a significantly higher rate than both S. paradoxus strains. Significance was assessed via chi-square tests with a Bonferroni correction. b) Sulfite mutation rates for haploids and diploids measured as the total number of non-escapee strains recovered over the total number of cells plated across all concentrations. Error bars represent the standard deviation of the mutation rate measurement assuming a Poisson distribution for colony counts. In both panels, mutagenized pool measurements are denoted as "MUT," and mock mutagenized pool measurements are denoted as "CON." Significance between species was assessed via chi-square tests, and "ns" denotes a nonsignificant difference.

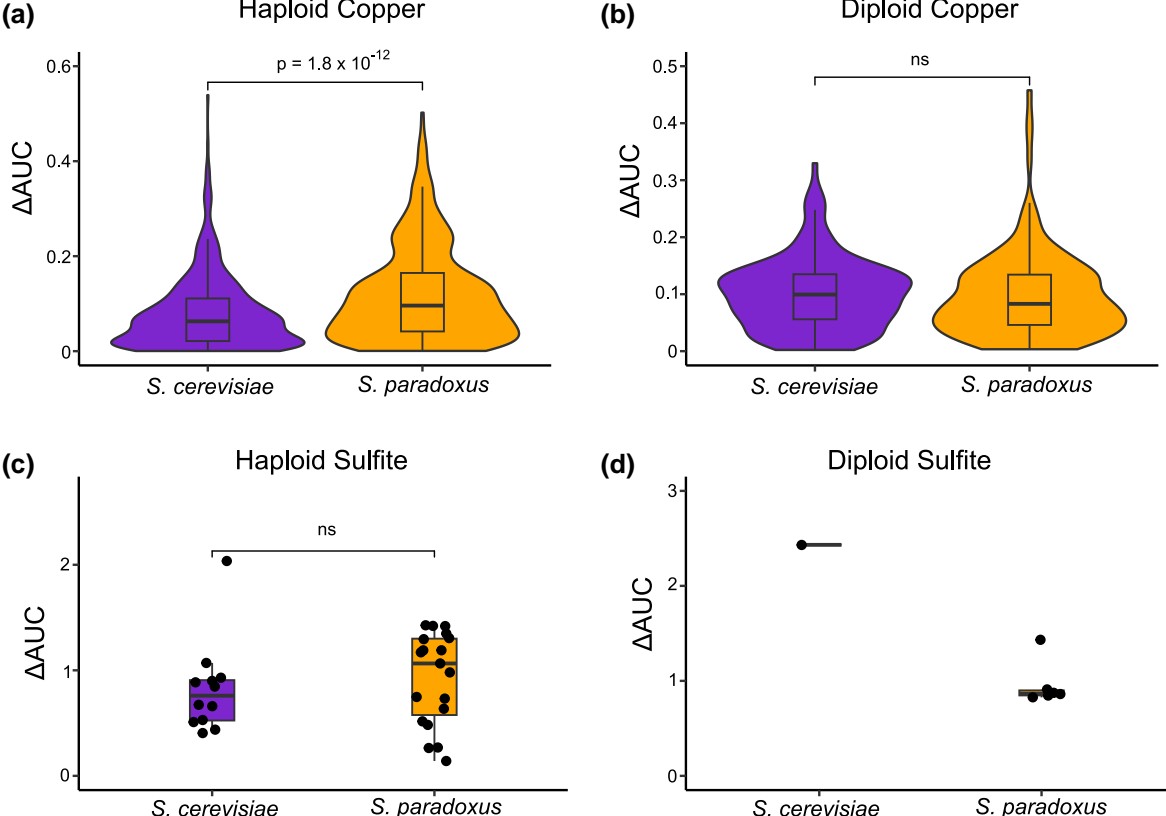

**Fig. 2.** Limited species differences in effect size for copper and sulfite resistance mutants. Resistance is measured by ΔAUC of colony size as a function of stressor concentration in mM. Significance values are derived from Kruskal–Wallis tests (ns indicates not significant). a) Haploid non-escapee copper mutants ($N = 1,107$) show a larger effect size in *S. paradoxus*. b) Diploid non-escapee copper mutants ($N = 309$) show no species difference in effect size. c) Haploid sulfite mutants ($N = 31$) show no difference in effect size. d) Diploid sulfite mutants ($N = 7$) have a low sample size that precluded a high-confidence comparison of effect size.

measured the change in AUC for colony size as a function of stressor concentration relative to the ancestor (ΔAUC) and only retained for analysis those mutants whose AUC was three standard deviations higher than the ancestral measurements to exclude physiological escapees. Following escapee exclusion, the final sample sizes for haploid and diploid copper mutants were 1,107 and 309 respectively. For sulfite mutants, these numbers were 31 and 7 respectively. These numbers represented a total retention of 75.6% of the copper phenotype data and only 2.0% of the sulfite phenotype data due to the high number of physiological escapees.

We find that haploid copper mutants of *S. paradoxus* have a larger average effect size than *S. cerevisiae* copper mutants (Fig. 2a). This result is driven by induced mutants (Kruskal–Wallis test, $P = 4.7 \times 10^{-12}$), and there is no species difference when only spontaneous mutants are considered (Kruskal–Wallis test, $P = 0.06$). In contrast, diploid copper mutants show no such species difference in effect size when spontaneous and induced mutants are considered together (Fig. 2b, $P = 0.25$) or when these groups are considered separately ($P = 0.08$ and $P = 0.12$ respectively, Kruskal–Wallis test). Of note, YJF5538 the S288c derivative used as a control, displayed the maximum AUC possible in this assay of 0.8, indicating its copper resistance is far greater than that of any of the recovered mutants. Recovery concentration was weakly though positively correlated with ΔAUC in haploid copper mutants (Pearson's $r = 0.20$, $P < 10^{-5}$), but not diploid copper mutants ($P = 0.26$).

For sulfite, we observed no difference in effect size between mutants of the 2 species among haploids (Fig. 2c, $P = 0.29$,

Kruskal–Wallis test). For diploids, we only recovered a single *S. cerevisiae* mutant that was not an escapee. Although this single ΔAUC measure was higher than the distribution of the six ΔAUC measures from the corresponding *S. paradoxus* mutants ($P < 2.2 \times 10^{-16}$, one sample t-test), these small sample sizes precluded high-confidence inference of an effect size distribution for diploid sulfite mutants in these species (Fig. 2d). Recovery concentration did not correlate with ΔAUC in haploid or diploid sulfite mutants ($P > 0.15$ in both cases). Overall, the phenotyping data do not support different mutational effect sizes in the 2 species as being a major driver of the natural pattern of differential adaptation seen between *S. cerevisiae* and *S. paradoxus*. The distributions of effect sizes between species are largely similar, and the one well-supported difference, haploid *S. paradoxus* exhibiting larger effect mutations for copper, does not align with the natural pattern of apparent constraints.

## Pleiotropic costs

To test whether resistance mutations in the 2 species tended to come with different pleiotropic costs, we also measured colony size on six permissive conditions for all mutants (Fig. 3). We measured costs as growth relative to the ancestor. For copper mutants, we find pervasive costs in both species and only a few instances of greater costs in *S. paradoxus* when the 2 species are compared (Fig. 3a and 3b, Kruskal–Wallis test, Bonferroni corrected, $P < 0.05$). For sulfite mutants, there were no differences in costs between species even before correcting for multiple comparisons (Kruskal–Wallis test, $P > 0.05$ in all cases).

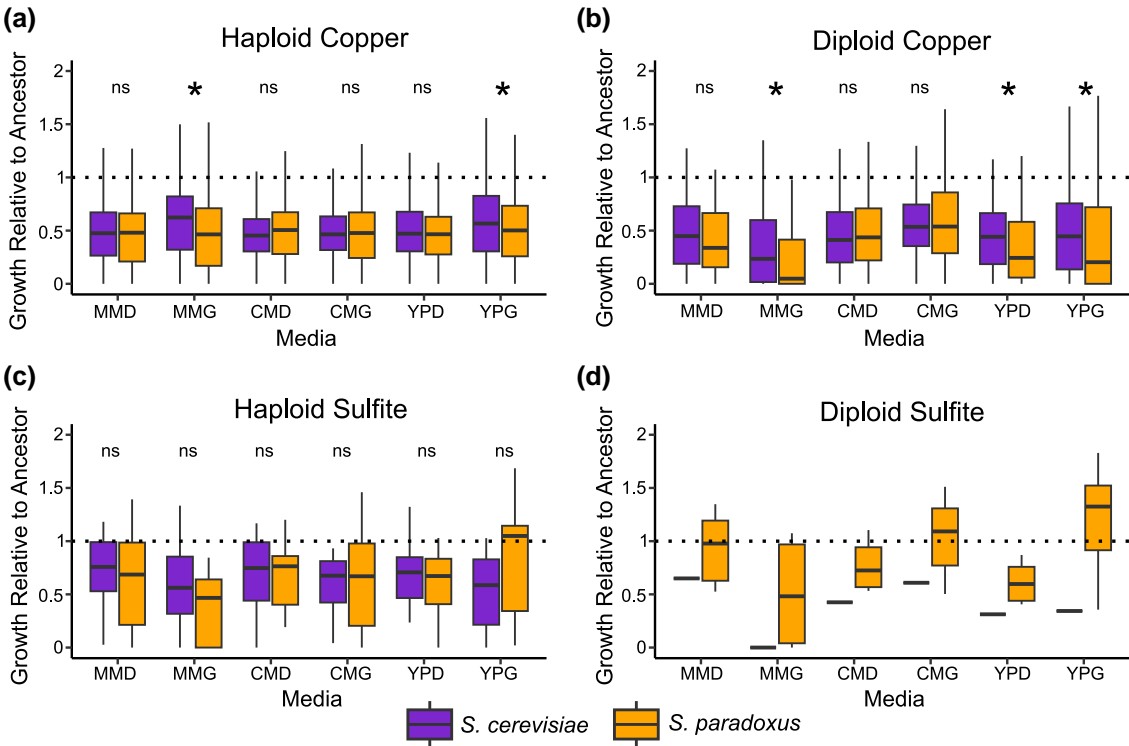

**Fig. 3.** Species differences in pleiotropic costs of recovered mutations. Costs are measured by growth (colony size) in six media relative to the ancestor. Nonsignificant differences are denoted by "ns," and significant differences are noted by an asterisk and represent $P < 0.05$ for a Kruskal–Wallis test after Bonferroni correcting for 6 comparisons. Media abbreviations are minimal media + dextrose (MMD), minimal media + glycerol (MMG), complete media + dextrose (CMD), complete media + glycerol (CMG), YP + dextrose (YPD), and YP + glycerol (YPG). a) Costs for non-escapee haploid copper mutants with data in all permissive conditions ($N = 999$). *S. paradoxus* mutants display greater costs in MMG and YPG. b) Costs for non-escapee diploid copper mutants with data in all permissive conditions ($N = 286$). *S. paradoxus* mutants display greater costs in MMG, YPD, and YPG. c) Costs for non-escapee haploid sulfite mutants with data in all six conditions ($N = 30$). There are no significant differences in costs. d) Costs for the diploid sulfite mutants recovered in this study ($N = 7$).

Following this, we also investigated the general relationship between costs and effect size across all mutants. Haploid and diploid copper mutants of both species showed a negative correlation between ΔAUC and cost relative to their ancestor (Supplementary Table 11, Supplementary Fig. 5, Spearman rank correlation, $P < 10^{-5}$ in all cases). Thus, mutants with larger effects on copper resistance tended to incur greater costs in both species. There was no interaction between species and resistance in predicting costs for either ploidy state, indicating the relationship is similar for the 2 species (2-way ANOVA, $P > 0.30$ in both cases).

Haploid sulfite mutants showed a different pattern (Supplementary Table 11, Supplementary Fig. 5) with no relationship between effect size and growth relative to the ancestor in *S. cerevisiae* (Spearman rank correlation, $P = 0.19$) but a negative relationship in *S. paradoxus* (Spearman rank correlation, $P = 0.04$). For diploid sulfite mutants, there was no relationship between effect size and costs for the six diploid *S. paradoxus* mutants. These results indicate that copper resistance mutations in both species tend to be more costly as their effects increase and that this relationship is less consistent across species for sulfite.

## Copper and sulfite resistance are caused by mutations in a small number of genes

To determine which mutations underlie copper and sulfite resistance in *S. cerevisiae* and *S. paradoxus*, we subjected 150 mutants and our 4 ancestral strains to whole genome sequencing (Supplementary Table 3, $N = 119$ and $N = 31$ for copper and sulfite,

respectively). After filtering, we found an average of 6.85 SNPs and insertions/deletions (indels) in each mutant. We found 20 genes with multiple disruptive variants within our dataset. Of these, nine were significant when compared to a simulated null distribution after correcting for multiple comparisons (see Methods). Of the 9, the 3 most commonly mutated genes were *PMA1* (copper, $N = 42$), *REG1* (copper, $N = 11$), and *RTS1* (sulfite, $N = 9$). Whereas *PMA1* and *RTS1* were mutated in both species, *REG1* mutations were only found in *S. cerevisiae*. There was no statistical difference in the incidence of *PMA1* or *RTS1* mutants between species (chi-square test, $P > 0.20$ in both cases), but there was a difference in incidence between species for *REG1* mutations (chi-square test, $P = 0.0013$, Supplementary Table 7). Of the remaining genes with multiple mutations, several were annotated as dubious ORFs or telomeric, indicative of genotyping errors rather than their having a role in copper or sulfite resistance.

*PMA1* mutations were called in 22 *S. cerevisiae* copper mutants and 20 *S. paradoxus* copper mutants. There was also one additional strain (YJF4439) not assigned a causal variant that harbored no coding changes in *PMA1* but had 6 synonymous changes in this gene. Among the 42 mutants with coding changes, there were six cases of site-level parallel changes between species and there were no *PMA1* mutations detected among the sequenced sulfite mutants. This high level of incidence and parallelism shows that mutations in this gene cause elevated copper resistance in both species. It also lends support to the estimate of the saturation of the screen being greater than one mutation per site. Of the 11 mutations found in *REG1*, 8 were nonsense mutations and 2

were frameshift mutations, signaling that loss of function REG1 mutations confer copper resistance in S. cerevisiae.

Among sulfite mutants, there was only one gene with a significant number of mutations: RTS1. This gene was mutated in five S. cerevisiae and four S. paradoxus sulfite mutants. Of these mutations, 5 were nonsense mutations, 2 were frameshifts, and 2 were missense mutations. These data indicate that loss of function mutations in RTS1 in both species confer sulfite resistance. In addition to this gene, there was another gene that did not meet our significance criteria but showed a high degree of interspecies parallelism among the sulfite mutants, signaling that it is likely causal: KSP1. This gene is mutated in three sulfite-resistant mutants of both species, representing a significant hit if the S. cerevisiae and S. paradoxus screens are considered together ($P = 5.2 \times 10^{-5}$, Bonferroni corrected). Of these 6 mutations, 5 are nonsense mutations, again signaling that loss of function mutations in this gene confer sulfite resistance in both species. Upon closer inspection of these mutants, it is likely that the three S. paradoxus mutants have a single origin, with each strain having exactly the same mutation and having been recovered from the same plate (Supplementary Table 3).

For mutations in PMA1, REG1, RTS1, and KSP1, we investigated their location and frequency across the gene body (Supplementary Figs. 6 and 7). We find that PMA1 mutations in both species are dispersed broadly across the gene with more mutations being found at the C-terminus than the N-terminus. REG1 mutations showed a strong N-terminal bias including a nonsense mutation at Lys12, indicating some of these mutations likely result in null alleles. For RTS1, recovered mutations were dispersed across the gene body, and for KSP1, we saw N-terminal nonsense mutations in S. cerevisiae, with one S. paradoxus mutation being found near the center of the gene body.

## Aneuploidy and CUP1 duplication cause copper resistance

Among the 119 copper mutants we sequenced, we found 45 strains that were aneuploid, nine of which were S. cerevisiae and 36 of which were S. paradoxus. Among copper mutants, almost every instance of aneuploidy either involved chromosome VIII, chromosome III, or both (Supplementary Table 3). Based on a simulated null distribution, aneuploidy of chromosome VIII is enriched in both species ($P < 5 \times 10^{-6}$ in both cases) and aneuploidy of chromosome III is enriched in S. paradoxus ($P < 10^{-7}$) but not S. cerevisiae ($P = 0.36$). This indicates that chromosome VIII aneuploidy is contributing to copper resistance in both species, and chromosome III aneuploidy is contributing to copper resistance in S. paradoxus. Although several S. paradoxus and S. cerevisiae strains have extra copies of both chromosome VIII and III, the frequency of this combination did not suggest an excess of co-occurrence given the frequencies of each respective aneuploidy ($P > 0.5$ in both cases). Comparing the 2 species, the incidence of aneuploidy of any kind, along with aneuploidy of either chromosomes III or VIII, was all significantly higher in S. paradoxus copper mutants (chi-square test, $P < 0.01$ in all cases, Supplementary Table 7). The CUP1 gene resides on chromosome VIII and likely explains aneuploidy-based copper resistance.

We also called structural variants in our sequenced strains using Delly. We only found 5 cases of high-confidence structural variants in our sequenced strains, and none could be assigned as likely causal for the phenotype (Supplementary Table 3, Supplementary Table 5). Many prior studies have seen that expansions of CUP1 tandem arrays confer greater copper resistance (Fogel et al. 1983; Adamo et al. 2012; Gerstein et al. 2015). To ensure

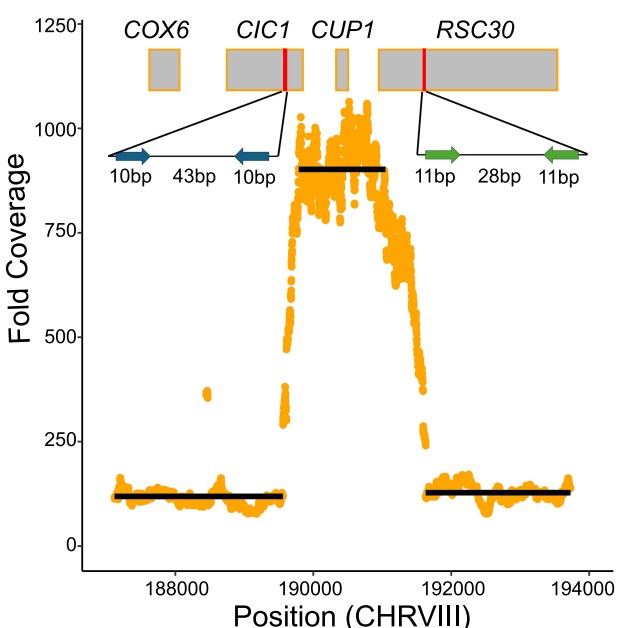

**Fig. 4.** A de novo CUP1 duplication occurred in an S. paradoxus mutant. The plot shows fold coverage as a function of chromosome VIII position for the S. paradoxus copper mutant YJF4464. Each point is a single nucleotide in the reference genome. Black lines represent average coverage across the positions they span. Rectangles are positions of genes plotted to scale. Coverage in this strain supports approximately 9 copies of CUP1. Red vertical segments represent two regions containing short, inverted repeats in the S. paradoxus genome. The insets are schematic depictions of these regions with the lengths of the repeats and intervening sequences. Inset colors indicate that the two regions contain repeats that are not identical in sequence to one another. Insets are not drawn to scale.

CUP1 duplications were not missed, we manually inspected coverage at the CUP1 locus for each sequenced copper-resistant strain. We found evidence of duplication of this gene in one S. paradoxus strain, YJF4464, with coverage data supporting 9 CUP1 copies (Fig. 4). Following mapping to a modified reference with an inverted repeat of the CUP1 region, we find reads that span the repeat junction and support an inverted-repeat structure. The duplicated region also is flanked on either side by 2 pairs of short, inverted repeats, suggesting the possibility of Origin-Dependent Inverted-Repeat Amplification (ODIRA) as the mechanism of this mutation (Brewer et al. 2011). We also manually inspected the SSU1 locus in the sulfite mutants and found no anomalous patterns in any of the strains.

## Significant variants: effect size and costs

We compared the effect size and costs for all of the inferred causal variants. We also performed these comparisons for the subset of strains that lacked any of the above mutations that were assigned as causal. These strains were subsequently designated as "unknown causal variant" in these analyses. When comparing effect size for similar mutations between species (Fig. 5a and 5b), we find that there is only one significant difference. Mutations in RTS1 tended to be of larger effect in S. paradoxus (Kruskal–Wallis test, $P = 0.014$). All other comparisons of effect size including mutants with unknown causal variants were not significant (Kruskal–Wallis test, $P > 0.08$ in all cases). We find that the CUP1 duplication strain has a ΔAUC of 0.256, placing it among the more resistant mutants. When costs were compared within mutant classes (Fig. 5c and 5d), we found that average costs relative to the ancestor in the six permissive conditions only significantly differed

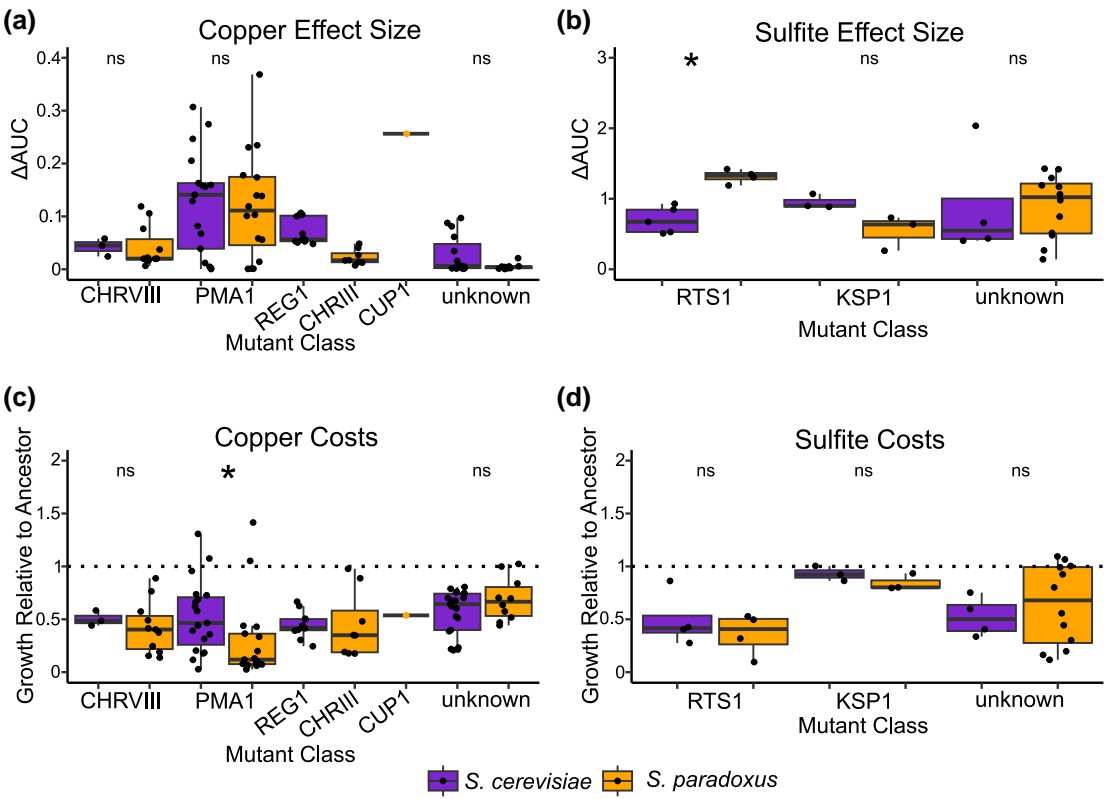

**Fig. 5.** Effect sizes and costs for the mutant classes assigned as causal. Effect sizes are measured as ΔAUC for the colony size measurements as a function of stressor concentration in mM. Costs are measured as average growth relative to the ancestor across six permissive conditions. Significant species differences are noted with an asterisk and represent $P < 0.05$ for a Kruskal–Wallis test after a Bonferroni correction, and "ns" denotes a nonsignificant difference. a) Copper effect sizes for the major copper mutant classes recovered for both species. There are no significant species differences. b) Sulfite effect sizes for the major sulfite mutant classes recovered for both species. *RTS1* mutants have a significantly higher effect size in *S. paradoxus*. c) Costs for the major copper mutant classes. *PMA1* mutants incur greater costs in *S. paradoxus*. d) Costs for the major sulfite mutant classes. There are no significant species differences.

between species for *PMA1* mutants such that *S. cerevisiae PMA1* mutants incur less severe costs than their *S. paradoxus* counterparts (Kruskal–Wallis test, $P = 0.01$). The *CUP1* duplication strain incurred an average cost relative to its ancestor of 0.54 across the six permissive conditions, indicating substantial costs.

## The PMA1 copper resistance mutations are partially recessive loss of function mutations

*PMA1* encodes a membrane-bound P2-type ATPase that exports protons from the cell to regulate intracellular pH and to establish an electrochemical gradient for secondary active transport into the cell (Young *et al.* 2023). Loss of function mutations in *PMA1* cause growth defects, pH sensitivity, sensitivity to cationic drugs, and have variable dominance (Morsomme *et al.* 2000; Cyert and Philpott 2013). If a weaker proton gradient mediates copper resistance in *PMA1* mutants, we expect our mutants to show pH sensitivity, a hallmark of loss of function mutations in *PMA1* (Cyert and Philpott 2013). By plating on low pH MM (pH = 2.5) we found that 27 of the 39 mutants tested showed a clear low pH sensitivity phenotype (Supplementary Table 8). When sensitivity is compared with copper resistance, we find that for both species low pH-sensitive *PMA1* mutants are more copper resistant than mutants that are not sensitive to low pH media (Kruskal–Wallis test, $P < 0.007$ in both cases). This indicates that the greater the magnitude of loss of function in Pma1, the greater the copper resistance.

To investigate the dominance of *PMA1* mutations, we backcrossed 5 pairs of mutants with site-level parallel changes

between species to their ancestors yielding 10 heterozygous diploid strains (Supplementary Table 9). We found that in 7/10 cases, heterozygotes displayed intermediate sensitivity to low pH MM, and we found in 10/10 cases, the heterozygotes had intermediate copper resistance compared to their ancestors. In contrast, 8/10 diploids showed little or no cost on CM plates (Supplementary Table 9). The results for the mutation Ala506Val are shown in Fig. 6 for both species. This mutation displays incomplete dominance regarding low pH sensitivity and copper resistance in *S. cerevisiae* and *S. paradoxus*. These data point to a general pattern of incomplete dominance for *PMA1* mutations in both species.

## The REG1 deletion phenotype is species-dependent

The recovery of *REG1* mutations in *S. cerevisiae* but not *S. paradoxus* led us to hypothesize that null alleles of *REG1* have a copper resistance phenotype in *S. cerevisiae* but not *S. paradoxus*. To test this, we deleted *REG1* from one of the *S. cerevisiae* ancestral strains (YJF3732) and one of the *S. paradoxus* ancestral strains (YJF3734) to determine if the effect of inactivating *REG1* varied between species. The *S. paradoxus* deletion mutant had an extremely slow growth phenotype upon isolation. As such, we also isolated a suppressor mutant derived from this deletion strain to include in our phenotype assay, and we included a 10× concentrate of the original mutant in the assay as well (Fig. 7a and 7b). We found that deletion of *REG1* in *S. cerevisiae* has a small effect on growth for CM and MM. However, in *S. paradoxus*, there is a strong growth

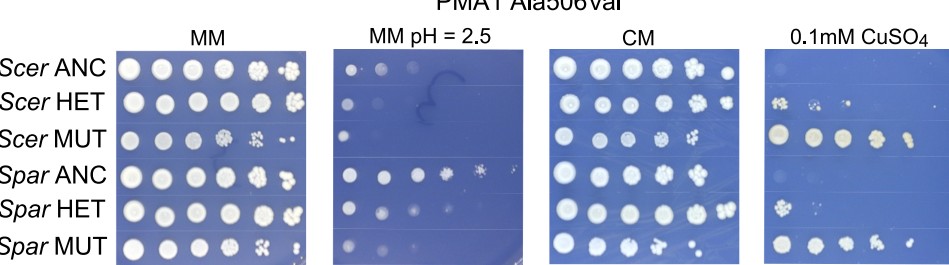

**Fig. 6.** Spot dilution phenotypes associated with the Ala506Val mutation in *PMA1* in *S. cerevisiae* and *S. paradoxus*. *S. cerevisiae* is denoted as "*Scer,*" and *S. paradoxus* is denoted as "*Spar.*" Haploid ancestors (ANC) are tolerant of low pH media and sensitive to 0.1 mM CuSO₄. Backcrossed heterozygotes (HET) show intermediate low pH tolerance and copper resistance compared to their ancestors and the haploid mutants. Haploid mutants (MUT) display low pH sensitivity and copper resistance.

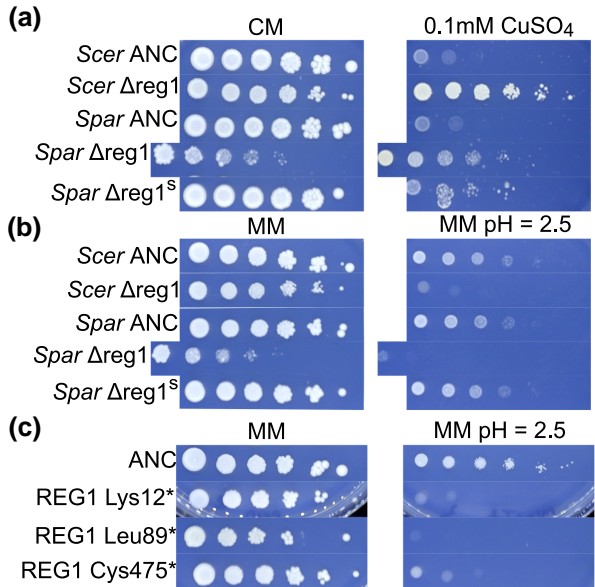

**Fig. 7.** Spot dilution phenotypes associated with *REG1* loss of function in both species. a) *S. cerevisiae* strain ancestor (YJF3732) and *S. paradoxus* strain ancestor (YJF3734) along with *REG1* deletion strains derived from these strains. *Spar* Δreg1ˢ refers to a suppressor mutant derived from "*Spar* Δreg1*.*" Deletion of *REG1* leads to a severe growth defect in *S. paradoxus* and confers elevated copper resistance in both species. b) Deletion of *REG1* leads to low pH sensitivity in *S. cerevisiae* and *S. paradoxus*. However, the suppressor mutation rescues wild-type growth. c) *S. cerevisiae* strain YJF3732 (ANC) and 3 *REG1* nonsense mutants identified among sequenced strains. All 3 mutants show low pH sensitivity.

defect in these permissive conditions. Wild-type growth rates in nonstress conditions were restored in the *S. paradoxus* suppressor mutant. For both species, deletion of *REG1* leads to an increase in copper tolerance, though this gain appears to be slightly greater in *S. cerevisiae*. These data are consistent with diverged mutational effects at the *REG1* locus between species and suggest that *REG1* mutations may not be a viable route to copper adaptation in *S. paradoxus* despite an effect on copper tolerance being conserved.

Prior literature has drawn functional connections between *REG1* and *PMA1* through the action of Reg1 on the Glc7 complex, which can play a regulatory role in Pma1's activity (Williams-Hart *et al.* 2002; Cyert and Philpott 2013; Mazón *et al.* 2015; Guarini *et al.* 2024). This led us to hypothesize that *REG1*'s effect on copper resistance in *S. cerevisiae* may be mediated through an effect of lowering Pma1 activity. If true, then *REG1* inactivation might be expected to yield a low pH-sensitive phenotype.

We spot-diluted several *REG1* nonsense mutants onto low pH MM and found that the *REG1* nonsense mutants we assayed have a low pH-sensitive phenotype, phenocopying *PMA1* mutants (Fig. 7c). This is consistent with the hypothesis that the copper resistance effect of these mutations may be mediated through an effect on Pma1. We also found a low pH-sensitive phenotype in both deletion strains but not the *S. paradoxus* suppressor mutant (Fig. 7b). These results provide evidence that *REG1*'s effect on copper tolerance may be mediated through an effect on Pma1 activity.

## Discussion

In nature, *S. cerevisiae* and *S. paradoxus* show contrasting patterns of adaptation regarding the anthropogenic enological stressors copper and sulfite. Although both species are present in vineyards, only *S. cerevisiae* has adapted to these chemicals, and it has done so multiple times. In this study, we investigated divergence in the DME for copper and sulfite resistance. Overall, our results show broadly conserved mutational effects along with high levels of gene-level and even site-level parallelism between species. However, we also found the subsets of mutations conferring resistance were not identical for the two species and that mutations in homologous genes could display species-specific effect sizes and costs. Broadly, these results do not provide a clear explanation for the absence of copper and sulfite resistance in *S. paradoxus*, but they highlight the presence of differences in the DME between species and some of the limitations of experimental characterization of the DME.

### Similarities and differences in the DME

In our first set of experiments, we found that the mutational target size for copper and sulfite resistance was similar between species. However, we did observe higher mutation rates at lower copper concentrations for *S. cerevisiae* than for *S. paradoxus*. This may reflect a true species difference, but the potential confounding effects of very slight differences in basal resistance and the greater chance of physiological escape at low concentrations make the interpretation of this result less clear. For sulfite, our assessment of the mutational target size differences was limited by the large number of physiological escapees. There are several potential explanations for this including stochastic variation in expression of *SSU1*. That said, the most plausible explanation is likely technical rather than biological. Sulfite plates have long been known to be difficult to work with due to off-gassing of SO₂ (Park *et al.* 1999), which can lead to variable sulfite concentrations batch to batch, and perhaps even within individual plates.

After we screened for mutants, we phenotyped a large number of them and assayed their effect size. Distributions of effect sizes for *S. cerevisiae* and *S. paradoxus* were largely similar. There was one significant difference between species, which was that haploid *S. paradoxus* copper mutants had a larger average effect size than their *S. cerevisiae* counterparts. In the context of the sequencing results, there is not an obvious explanation for the difference between haploid copper mutants of the two species. Within the main classes of copper mutants we observed (chromosome VIII aneuploids and *PMA1* mutations), we did not see a difference in effect size between species. However, this subset is biased for the most common types of mutations that confer copper resistance. Thus, it appears likely that rarer mutant classes that we were unable to detect in our sample underlie the species difference in effect size we see for haploid copper mutants.

Additionally, the species difference for copper does not persist in diploids, signaling the possibility of differences in the dominance of copper resistance mutations between species. However, when we backcrossed *PMA1* mutants to the ancestral strain, they showed a consistent pattern of incomplete dominance in both species, signaling that differences in dominance for *PMA1* mutations likely do not explain the difference between the haploid and diploid copper results. Variable effects of aneuploidy on haploids and diploids are expected due to differences in the relative increase in gene dosage and could thus account for some of the effect size differences between haploids and diploids.

Regarding sulfite, effect sizes were not different between species. However, in the context of the sequencing results, mutations in *RTS1* were present in both species, but their effect sizes differed significantly. This illustrates an interesting property that can be displayed by the DME: Although the DME can overall be quite similar between species, there can still be meaningful gene-level idiosyncrasies in effect size despite these general similarities.

Our investigation of pleiotropic costs showed that *S. paradoxus* mutants with elevated copper resistance tended to incur greater costs than their *S. cerevisiae* counterparts in several permissive conditions for both haploids and diploids. We found that this difference is not explained by differences in costs between strains aneuploid for chromosome VIII. In contrast, we do observe significantly greater costs in *S. paradoxus PMA1* mutants compared to their *S. cerevisiae* counterparts. This common mutant class may therefore partially explain the few differences in costs we do observe. Similar to effect size, the pattern may also in part be driven by rarer unidentified mutant classes.

Taken at face value, the presence of greater costs in *S. paradoxus* in a few conditions offers an attractive potential explanation for apparent differences in the adaptive capacity for copper. In principle, if such a difference were meaningful in nature it could increase the waiting time for a second beneficial mutation to occur before local extirpation (Orr and Unckless 2014). However, if rare, costless mutations are the most relevant to adaptation, then a species difference in costs (among costly mutations) may not be ecologically relevant because these mutations never persist in nature for either species. Without a class of resistance mutations that are costly in *S. paradoxus* but not in *S. cerevisiae*, costs do not offer a clear explanation for natural outcomes.

## Identity and effects of sequenced mutants

Among our sequenced strains, we were able to identify many causal variants underlying copper and sulfite resistance in both species. For copper, our screen yielded a large number of strains harboring mutations in the gene *PMA1*, which encodes an essential membrane-bound proton efflux pump. Pma1 is biochemically well characterized (Morsomme *et al.* 2000; Young *et al.* 2023) and has previously been implicated in copper resistance (Gerstein *et al.* 2015). The most plausible mechanism for this effect based on our experiments appears to be reduced copper influx due to a weaker electrochemical gradient (Morsomme *et al.* 2000; Cyert and Philpott 2013). However, the ecological relevance of these mutations is highly questionable. Wine fermentations are typically very acidic environments and proper Pma1 function can be rate limiting for growth (Cyert and Philpott 2013; Williams *et al.* 2015). On these grounds, we speculate that the costs of *PMA1* mutations are likely far too detrimental to allow for these mutations to persist in vineyards or contribute to domestication-associated copper resistance.

The other major class of copper mutations shared by the 2 species was aneuploidy of chromosome VIII. For *S. paradoxus*, aneuploidy of chromosome III was also significant, but the reason for this phenotype is not obvious because none of the major genes implicated in copper resistance in yeast such as *CUP1*, *CUP2*, *CRS5*, and several others reside on this chromosome (Culotta et al. 1994; van Bakel *et al.* 2005; Gerstein *et al.* 2012; Caudy *et al.* 2013; Chang *et al.* 2013). In contrast, the explanation for the phenotype of chromosome VIII aneuploidy is easily accounted for because the *CUP1* gene resides on chromosome VIII in both species. This aneuploidy has been known to cause copper resistance for many decades (Fogel *et al.* 1983) and was also seen in the study by Gerstein *et al.* (2015). Aneuploidy is caused by nondisjunction and occurs at appreciable rates in *S. cerevisiae* with higher rates for smaller chromosomes (Gilchrist and Stelkens 2019). It has been argued that aneuploidy represents an excellent avenue for short-term adaptation to drastic changes in the environment because it often represents an accessible mutational path that can have large effects due to its increase in gene dosage for the aneuploid chromosome (Gerstein and Berman 2015). Aneuploidy has also been hypothesized to be valuable to adaptation because it is reversible, and this has garnered some empirical support (Yona *et al.* 2012). In this way, aneuploidy can be seen as a short-term adaptive solution that increases the potential waiting time for a rarer adaptive variant to arise.

Chromosome VIII aneuploidy is of particular interest for copper resistance due to how it may impact the rate *CUP1* tandem expansion. As pointed out by Zhao *et al.* (2014) and Fogel *et al.* (1983), tandem expansion of a multi-copy *CUP1* array is a fundamentally different process and far more common than de novo formation of a tandem array from a single copy. Many studies have observed the expansion of *CUP1* arrays that were already multi-copy, and this occurs via unequal crossovers (Fogel and Welch 1982; Fogel *et al.* 1983; Adamo *et al.* 2012; Gerstein *et al.* 2015; Hull *et al.* 2017; Zhao *et al.* 2017). However, to our knowledge, no prior study has explicitly observed de novo formation of a *CUP1* array from a strain that began with a single copy until now. Fogel *et al.* (1983) posited that aneuploidy of chromosome VIII may potentiate *CUP1* tandem expansion. This would align with the proposed mechanism of how *CUP1* tandem arrays originate, which requires two simultaneous double-strand breaks on separate chromosomes and erroneous non-homologous end joining of the 2 chromosomes (Zhao *et al.* 2014). However, we did not observe any tandem duplications, and the repeat structure we observed is better explained by a different model.

Upon close inspection of the *CUP1* duplication in *S. paradoxus*, we found evidence for an inverted-repeat structure similar to a *SUL1* inverted-repeat structure described by Araya *et al.* (2010). As elaborated by Brewer *et al.* (2011), the Origin-Dependent Inverted-Repeat Amplification (ODIRA) model offers a potential

explanation for how these structures can arise. The model requires a nearby origin of replication and nearby flanking short, inverted repeats. Consistent with this model, the *CUP1* locus is near the *ARS810* locus in *S. cerevisiae*, and in *S. paradoxus,* there exists a pair of short (10–11 bp), inverted repeats directly upstream and downstream of the duplicated region we observe. Only *S. paradoxus* harbors inverted repeats of 10 bp or more on either side of the *CUP1* region, but smaller inverted repeats (7 bp or fewer) are highly abundant in both species on either side of the gene. The coverage pattern also increases and decreases in a stepwise pattern, similar to the CNVs described by Todd and Selmecki (2020).

If this duplication arose via this mechanism, it would raise many questions about the origin of *CUP1* duplications. For instance, why have these structures not arisen in *S. cerevisiae*? One potential explanation is that *S. cerevisiae* lacks inverted repeats of sufficient length at this locus to drive the ODIRA mechanism. Irrespective of the explanation, our data demonstrate evidence for a different mechanism of *CUP1* amplification in *S. paradoxus* than what has been documented in *S. cerevisiae*.

The final set of copper resistance mutations we identified were loss of function mutations in *REG1* in *S. cerevisiae*, but we did not observe any mutations in *REG1* in *S. paradoxus*. We found that deletion of *REG1* in *S. paradoxus* led to an extreme slow growth phenotype and increased copper resistance, explaining the absence of *S. paradoxus REG1* mutants in our screen. Reg1 is a regulatory subunit of the Glc7 complex which can play a role in regulating Pma1 activity (Williams-Hart *et al.* 2002; Cyert and Philpott 2013; Mazón *et al.* 2015; Guarini *et al.* 2024), and we found *REG1* mutants have a low pH sensitivity phenotype. This is consistent with loss of function mutations in *REG1* leading to lower Pma1 activity. Curiously, the idea that lack of *REG1* may lower Pma1 activity appears to be overtly contradicted in a biochemical assay of Pma1 activity in a *REG1* deletion background (Young *et al.* 2010). Two other studies have also investigated Glc7's capacity to dephosphorylate Ser899, Ser911, and Thr912 of Pma1 in a *REG1* deletion background and found these functions are not dependent on the presence of *REG1* (Mazón *et al.* 2015; Guarini *et al.* 2024). Thus, Reg1 and Pma1 appear to interact via the Glc7 complex, but the precise functional relationship between these gene products remains unclear. It is also unclear whether the higher cost of *PMA1* mutants in *S. paradoxus* is related to the higher cost of *REG1* deletions.

For sulfite, the main route to adaptation in domesticated *S. cerevisiae* strains has been via structural variants that alter the upstream sequence of *SSU1*. This type of mutation has occurred at least 3 times independently (Goto-Yamamoto *et al.* 1998; Pérez-Ortın *et al.* 2002; Zimmer *et al.* 2014; García-Ríos *et al.* 2019). In many prior studies *FZF1* and *SSU1* have been the principal genes presumed to make the largest contributions to sulfite tolerance (Park *et al.* 1999). Surprisingly, we did not observe any mutations in these genes in our dataset. Instead, we found two causal genes, *RTS1* and *KSP1*, that have effects on sulfite tolerance in both species. *RTS1* encodes a regulatory subunit of protein phosphatase 2A, and it is thought to play several roles in mitosis (Shu *et al.* 1997). One interesting finding from Linderholm *et al.* (2008) points to a potential role in sulfite metabolism for *RTS1*. In this screen, the *RTS1* deletion mutant was found to be a hyperproducer of hydrogen sulfide, indicating greater sulfite reductase activity. In our data, we find that nonsense mutations in *RTS1* confer greater sulfite resistance. Taken together, these results suggest that null alleles of *RTS1* may exert an effect on sulfite tolerance via increasing the activity of sulfite reductase within cells. If true, this would represent a mechanistic departure from how

sulfite tolerance has been achieved in domesticated strains, which universally increase sulfite efflux.

The other gene in which we observed disruptive hits in sulfite mutants for both species was *KSP1*. This gene encodes a serine/threonine protein kinase involved in TOR (target of rapamycin) signaling and autophagy (Umekawa and Klionsky 2012). *KSP1* has not been explicitly implicated in sulfite tolerance to our knowledge. However, we speculate that this effect is likely mediated through its effect on autophagy. It was recently shown that autophagy is required for sulfite tolerance in *S. cerevisiae* (Valero *et al.* 2020), and Ksp1 is a negative regulator of autophagy (Umekawa and Klionsky 2012).

Overall, the choice to sequence only haploids may have introduced biases for loss of function mutations with some recessive character among our sequenced mutants given the causal mutations in our dataset. These mutations may be of limited ecological relevance due to most natural isolates being diploid. That said, any large-effect dominant mutation would also be accessible in haploids, and *PMA1* mutations were shown to have an effect on copper resistance in diploids. However, sequencing diploid copper and sulfite mutants would likely offer additional insight into the genetics and evolution of these two traits.

## Implications for copper and sulfite adaptation in nature

Natural domesticated isolates of *S. cerevisiae* that have adapted to resist copper and sulfite have done so via known genetic mechanisms. The absence of these large-effect mutations in *S. paradoxus* could be driven by many factors including differences in population size in vineyards, niche differences such as contrasting in source/sink dynamics in vineyards, or differential mutational access (Holt and Gaines 1992). These known mutations are of extremely large effect and would be easy to detect. None of our recovered mutants had copper resistance approaching that of YJF5538, an S288c derivative harboring multiple copies of *CUP1*. Additionally, based on the phenotypes obtained in prior studies, none of our sulfite mutants are as resistant as domesticated strains harboring rearrangements (Pérez-Ortın *et al.* 2002; García-Ríos *et al.* 2019). These comparisons support that large-effect structural variations were not erroneously missed in our sample of sequenced strains. The apparent rarity of these mutations coupled with the fact that they have occurred in parallel in *S. cerevisiae* point to a strong likelihood that (short-term) effective population sizes for this species in vineyards are extremely large (Karasov *et al.* 2010). However, *S. paradoxus* is likely comparably abundant in vineyards (Dashko *et al.* 2016). This raises a critical issue concerning studies of the DME in relation to adaptive outcomes in nature. Namely, extremely rare mutations that are beyond the detection limit of normal screens can play important roles in adaptation, especially for organisms with large population sizes like microbes. Measures of the DME may therefore be meaningfully biased for common mutations even if the screen is "saturated" for point mutations as ours was.

Although seemingly quite rare, we did recover a single strain with a mutation similar to those observed in domestication strains. The strain in question is an *S. paradoxus* strain that harbors approximately 9 copies of *CUP1*. This strain is fairly copper-resistant and incurred consistent costs across the permissive conditions tested. These results are of interest because this strain has a modest copper resistance phenotype compared to YJF5538 and other *S. cerevisiae* strains with multiple copies of *CUP1* (Strope *et al.* 2015). This raises the possibility of differential effect size of *CUP1* mutations in the different species, but further investigation is needed especially

considering the inverted-repeat structure observed. Additionally, this strain's persistent costs are inconsistent with what is known about copper-resistant strains of *S. cerevisiae* harboring *CUP1* duplications, which have no apparent growth defect. As such, this result similarly raises the possibility of species-specific costs to *CUP1* amplification in *S. paradoxus*, but more detailed study is also needed.

These results raise many other questions about *CUP1* tandem arrays more generally. Firstly, it is unclear why duplication of this gene appears to be much rarer than for other genes such as *SUL1* (Sanchez *et al.* 2017). Close inspections of the known *CUP1* tandem structures have shown that the two genes flanking *CUP1*, *CIC1* and *RSC30*, are never included intact in the tandem duplications, which Zhao *et al.* (2014) noted as potentially indicating that extra doses of these genes are poorly tolerated. *CIC1* is also essential, meaning any *CUP1* duplication must keep a functional copy of *CIC1* intact. The *CUP1* duplicated region in *S. paradoxus* also only covers part of the *CIC1* and *RSC30* genes, similar to *CUP1* duplications in *S. cerevisiae*. These potential constraints based on genomic location may explain why this duplication seems to be far rarer than duplication of *SUL1* and other genes. Whether these species differ in the rate of this mutation or if the ecology of these species in vineyards differs more than is currently appreciated both remain open questions. What is clear is that *CUP1* duplications are relatively rare and may occur at a rate near the detection limit of the screen performed in this study. We screened ∼$10^8$ cells in total and recovered a single case of de novo *CUP1* duplication. Given these mutations are unlikely to be induced via UV exposure, the mutation rate for *CUP1* amplification may be on the order of $10^{-8}$ or lower.

Among the sulfite mutants we sequenced, we did not observe any rearrangements involving *SSU1*, meaning the rate of (spontaneous) occurrence of these mutations is lower than the detection limit of the screen we performed. Translocations and inversions are relatively rare compared to other types of mutations, and it is conspicuous that these rare classes of mutants have repeatedly been selected for in sulfite-exposed strains. This suggests that very few mutations of large effect are available to *S. cerevisiae* with respect to the acquisition of greater sulfite resistance. Complicating the matter further, CRISPR induction of one of the translocations showed that this mutation lowers sulfite tolerance in a background lacking duplications of a 76 bp repeat in the *ECM34* promoter, demonstrating background-dependent effects of this mutation (Pérez-Ortín *et al.* 2002; Fleiss *et al.* 2019).

Overall, these observations are consistent with the extremely small mutational target size we observed for sulfite in our mutant screen. It is likely that very few mutations in the genome aside from extremely rare rearrangements can elevate *SSU1* expression to the levels needed for success in a winemaking environment. This raises questions about the regulation of *SSU1* and the mutations available in its promoter sequence and its primary transcription factor, Fzf1, namely, why is greater *SSU1* expression seemingly so mutationally inaccessible? The ecological relevance of *FZF1* and *SSU1* to these species is further evidenced by the presence of introgressions of these genes from *S. paradoxus* into *S. cerevisiae* in the Mediterranean oak lineage but not the European wine lineage (Almeida *et al.* 2017). The functional and ecological significance of these different alleles in vineyard vs oak forest environments remains an open question.

Taken together, the results of this study offer new insights into the target size, effect size, costs, and genetic identity of mutations conferring copper and sulfite resistance in *S. cerevisiae* and *S. paradoxus*. They also raise many important questions concerning the ecology of these species, and the explanatory and predictive utility of assaying the DME.

## Data availability

Raw reads produced in this study were deposited into NCBI under BioProject PRJNA1107929. The 3 colony size datasets used in this study (haploid, diploid, and sequenced) along with all of the accompanying metadata can be found at DOI: 10.6084/m9.figshare.25777512. This DOI also includes all images used for data collection of colony size and the processed data used to generate the figures.

Supplemental material available at GENETICS online.

## Acknowledgments

The authors would like to acknowledge Douda Bensasson for providing the natural isolates used to generate the focal strains used in this study. The authors would also like to acknowledge James Miller and other members of the Fay lab for helpful and thorough feedback on the writing of this work. Also, the authors acknowledge and thank Elaine Sia for allowing access to her microscope for the tetrad dissections performed in this study. The authors would like to thank Nancy Chen, Elizabeth Grayhack, and Allen Orr for their feedback and suggestions on this work. Lastly, the authors would like to thank the 3 anonymous reviewers and the handling editor for constructive comments that improved the quality of this work greatly.

## Funding

This work was supported by National Institutes of Health Grant GM080669.

## Conflicts of interest

The author(s) declare no conflict of interest.

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

*Editor: S. Otto*