## [Peer Review File · Genetics]

The distribution of beneficial mutational effects between two sister yeast species poorly explains natural outcomes of vineyard adaptation

Emery Longan and Justin Fay

NOTE: The reviews and decision letters are unedited and appear as submitted by the reviewers.

In extremely rare instances and as determined by a Senior Editor or the EIC, portions of a review may be redacted. If a review is signed, the reviewer has agreed to no longer remain anonymous.

The review history appears in chronological order.

Review Timeline:

Submission Date:	2024-06-02
Editorial Decision:	2024-07-08
Resubmission Received:	2024-08-17
Editorial Decision:	2024-08-24
Revision Received:	2024-09-21
Accepted:	2024-09-23

July 8, 2024

GENETICS-2024-307078

The distribution of beneficial mutational effects between two sister yeast species poorly explains natural outcomes of vineyard adaptation

Dear Dr. Longan:

Two experts in the field have reviewed your manuscript, and I have read it as well. I am pleased to inform you that, with minor revisions, it is potentially suitable for publication in GENETICS. The reviewers have comments and concerns that need to be addressed in a revised manuscript. You can read their reviews at the end of this email.

It is most important that you address the following in your resubmission: streamlining and shortening the text. All reviewers thought that the article was long, with material that was interesting but could be removed or shortened without harming the main story (especially topics repeated in different sections). The reviewers all give suggestions for about how this can be done, but I leave it up to the authors to determine how best to streamline. In addition, the authors should at least discuss why haploids were chosen for sequencing when the strains tend to be diploid in nature (how might this bias conclusions) and how the methods used to accumulate mutations might have altered the types of mutations studied (e.g., duplications such as CUP might have been more often observed if mutagenesis was not a major source of new mutations).

We look forward to receiving your revised manuscript. Please let the editorial office know approximately how long you expect to need for revisions.

Upon resubmission, please include:

1. A clean version of your manuscript;
2. A marked version of your manuscript in which you highlight significant revisions carried out in response to the major points raised by the editor/reviewers (track changes is acceptable if preferred);
3. A detailed response to the editor's/reviewers' comments and to the concerns listed above. Please reference line numbers in this response to aid the editors.

Additionally, please ensure that your resubmission is formatted for GENETICS.

<https://academic.oup.com/genetics/pages/general-instructions>

Follow this link to submit the revised manuscript: Link Not Available

Sincerely,

Sarah Otto
Associate Editor
GENETICS

Approved by:
David Begun
Senior Editor
GENETICS

Reviewer #1 (Comments for the Authors (Required)):

The research demonstrates good scientific merit, the methods were quite adequate and robust. However, I have raised just a few issues bordered on clarity and structure.

Reviewer #2 (Comments for the Authors (Required)):

The premise of the study is that copper and sulfite resistance evolved in *S. cerevisiae* for winemaking, however, *S. paradoxus* in vineyards has not adapted in these conditions despite occupying the same environment and co-occurring with the sister species.

The authors posit different hypotheses to explain the limits of adaptation found in *S. paradoxus* compared to *S. cerevisiae* and measure several factors in resistance mutations: rate, effect size, pleiotropic cost. These hypotheses address interesting questions about the limits of adaptation (e.g. greater adaptive capacity of *cerevisiae*? is *paradoxus* genetically constrained?) and use sister species to test these hypotheses. When they evolved the strains, however, they found similar mutations in both species assayed, suggesting that mutational effects do not explain this difference found in nature. This makes me wonder if the copper sulfate addition in the winemaking process perhaps has more of an impact than previously thought? I understand that this has been looked into, but could there be some factors that have not been taken into account?

I think this is a notable contribution to the understanding of adaptation to toxic environments. I think the experiment is sound and interesting. The text needs clarification and editing throughout to focus the important information, plus organising the information so that it's easily accessible. At the moment there seems to be parallel stories happening and a more clear and explicit delineation of each one would help the reader follow them. I think the writing needs a lot of rearranging. I tried to give pointers where it seemed more important. Here are some examples:

>The Introduction should more organically connect the theoretical predictions for adaptive constraints and DMEs with the background of the history and genetics of copper and sulfite adaptations and tolerance.

>Perhaps include a bit more information about the statistical methods for each of the hypotheses and follow-up tests (e.g. spot dilution tests). E.g. L266-269 among strains at three concentrations is 1 paired t-test? Was that multiple tests?

>Focal strain set for this study included 12 strains derived from the four wild isolates. 150 mutants derived from those strains were sequenced -> total found in the methods, however, it would be helpful in other parts of the manuscript

>L491-496 are the detailed hypotheses from the introduction. Perhaps swap the formats between the intro and this? Quick reminder in the results and more detailed set-up in the introduction?

>In Table S3 please indicate columns of the superscript notation. e.g. 2 is column Q, 3 is column H. It wasn't easy to find as they are not unique enough to be searchable

>L239-243 these lines seem to a better fit with the following section where mutation rate measurements are explained.

>L497 It might be helpful to start/indicate a first subsection addressing the first hypothesis. Leave the short introductory paragraph with the summary and hypotheses.

>Check for gene nomenclature throughout. E.g. L723 names of genes should be fixed.

>I found "The REG1 deletion phenotype is species dependent" especially illuminating and interesting, although I had to read the section multiple times to understand how the experiment was set up to test this interaction. The relationship between REG1 and PMA1 seems very important in understanding the importance of genetic background. Perhaps edit to clarify the importance of this potential interaction?

>Figures: boxplots might work best as violin plots.

>The discussion should be re-written and shortened to follow more explicitly with the title of the subsections the different stories the paper is addressing. Additionally, some of the discussion could be included in the results (e.g. L1000-1003)

Reviewer #3 (Comments for the Authors (Required)):

I enjoyed thinking about the experiments in the manuscript "The distribution of beneficial mutational effects between two sister yeast species poorly explains natural outcomes of vineyard adaptation." The paper seeks to explain the observation that *Saccharomyces cerevisiae* has readily adapted to copper and sulfite in the context of vineyards, while the sympatric and closely related species *Saccharomyces paradoxus* has not. They acquired mutant haploid and diploid colonies on 8-12 levels of copper and sulfite from UV mutagenized and mock pools derived from two isolates from each species isolated off trees in Europe. They phenotypically screened nearly 4000 isolates and whole genome sequenced 150 haploids. They tested whether there are species-level differences in mutation rate, beneficial mutation effect sizes, or the pleiotropic costs of beneficial mutations in nonstressful conditions. Although some minor differences were reported, overall, they concluded that none of these factors could explain the observed adaptive outcome seen in vineyards.

My comments and questions are generally made to better understand how certain experimental decisions were made and whether analysis choices could have influenced the interpretation.

1. I had a hard time following how mutants were picked in the methods section, and some information seems to be missing. I

suspect a flow diagram might help to see how the provided numbers are linked to genotype and drug concentration they were isolated from; from the subset of 3024 haploids and 720 diploids that produce colonies, what ended up being sequenced? Figure S1A could be expanded to include numbers and more specifics to show how you go from two species x two backgrounds x 8-12 concentrations to the number of mutants selected for phenotyping and then sequencing.

2. I am curious whether mutants acquired at higher drug concentrations differ in mutation type or effect size. The text states that mutants were recovered from higher concentrations in *S. cerevisiae* than from *S. paradoxus*, but I didn't see anywhere else in the manuscript where this was directly interrogated. For the copper sequenced isolates, it was stated that they were evenly distributed across the four haploid ancestral strains - what was the split among drug concentration of isolation? There are a number of recent papers in *Candida albicans* showing that the mutations acquired at lower drug concentrations are different genotypically and phenotypically than the mutations at higher concentrations, and it would be interesting to see whether that is also true here.

3. For testing mutational target site, I wonder if collapsing all of the analysis to a paired t-test within drug, without accounting for the other variables (strain, concentration) has obscured some interesting patterns. Given that the underlying data is count data, a t-test is probably not the most accurate test, regardless. Based on Figure 1, it looks like *S. cerevisiae* has a higher mutation rate than *S. paradoxus*, and given that this result is central to the conclusions, I think it's really critical to ensure the statistics are done correctly.

4. The phenotypic assessment is dependent on the Δ AUC calculation, and I wonder whether some aspect of the analysis could be averaging out some of the variation among mutation effects.

- With respect to ancestral values - Were the ancestral AUC values variable? How many replicates of the ancestral strains were measured to ensure precision in the ancestral average AUC? Was the normalization procedure needed?

- It doesn't seem obvious from a visual assessment of Figure S2 (but the effect might be subtle). If you look at the technical replicates that were at different places in the plates, is the data consistent with an edge effect?

- How well does Δ AUC correlate with growth at one of the higher levels, such as 0.3 mM?

- I was confused by the statement that only mutations that were 3SD above the ancestor were included in downstream analysis. In Figures 2, 5, and S5 it looks like there are many mutations very close to 0 Δ AUC. As above, were the mutants within 3SD (termed by the authors as physiological escapees, but this could also be low s mutations?) most likely to come from low drug plates?

- If you look at the higher effect mutants, does anything jump out about where they came from?

5. Given that the paper's stated goal was to understand the observation that *S. cerevisiae* adapts more readily to copper and sulfite than *S. paradoxus* in nature, I had expected the diploids to be featured more prominently in the manuscript than haploids. I was very surprised to see that only the haploid copper mutants were sequenced, and I'm wondering why that decision was made. I do not suggest additional sequencing lightly, but it would be very interesting from a ploidy evolution perspective, and I think more ecologically relevant, to also look at the mutations that arose in the diploid strains.

6. Can you be more specific for the "unknown" mutations? Were there multiple well-supported variants in these lines?

7. The discussion talks about how the known mechanisms for copper and sulphite resistance in nature were largely not obtained through their screen. Is this due to the biases of UV-C mutagenesis, not just that these mutations are rare and hence 'beyond the detection limit of normal screens' (L972)? A discussion of the mutants that arose through mutagenesis vs. the mock would also be helpful (if some of the mock mutants were indeed sequenced).

8. Both the Introduction (over 6 pages) and Discussion (over 11 pages) are quite long and could be tightened up to focus on the things that matter most to the reader and to place the results in the context of the literature. There are essentially no references for the first several pages of the discussion, by example. Other parts of the discussion verge quite deep into the mechanistic basis of different types of mutations, which, although interesting to read about, were fairly far afield of the experiments that were done.

Associate Editor Comments:

In addition to the suggestions above, I found myself puzzled when reading the abstract that it focused so much on evolutionary constraints without mentioning ecological or phenotypic constraints. It could be, for example, that niche differences between *S. cerevisiae* and *S. paradoxus* make *S. cerevisiae* differentially predisposed to domestication (e.g., tendency to colonize vats of wine, pre-existing copper and sulfite tolerance, artificial selection favouring *S. cerevisiae*, e.g., because of the flavour of the wine). There are many reasons that one species may have repeatedly adapted to wine brewing in the presence of copper and sulfite, which do not relate to evolutionary adaptability.

The main text didn't raise this concern so much in my mind because it focused on a narrower question: do they have differential access to mutations? But the abstract seemed misleading, I believe, in claiming that the explanation for the preponderance of *S. cerevisiae* in copper/sulfite-adapted wine yeast is due to their greater ability to adapt: "...*S. paradoxus*, has not adapted to these chemicals despite being consistently present in sympatry with *S. cerevisiae* in vineyards. This contrast represents a case of apparent evolutionary constraints favoring greater adaptive capacity in *S. cerevisiae*."

Put another way, population size matters enormously for evolutionary adaptability, even if the target size in the genome and the distribution of fitness effects are the same. Thus, ecological differences could allow *S. cerevisiae* to adapt to conditions that *S. paradoxus* cannot, simply because they survive in high enough numbers to adapt.

The paper might benefit from a short discussion of the role of ecological differences and a rewrite of the abstract to better reflect that it is a null hypothesis to be tested that *S. cerevisiae* & *S. paradoxus* are equally able to adapt to copper/sulfite.

Reviewer 1 Comments

In this study, we UV mutagenized copper and sulfite sensitive strains of 177 *S. cerevisiae* and *S. paradoxus* and directly recovered mutants from plates containing various 178 concentrations of these chemicals to assay mutational target size [Line 176 - 178 , could you consider rephrasing the last part of the sentence for clarity].

Line 179... mutational effect size and pleiotropic cost. [Little explanation of the term may help readers have a better understanding].

Line 182-192 seems like a portion of methods, [do consider moving to the materials and methods section to maintain structure and reasonable length of the introductory section].

Line 183do not align with the pattern seen in nature. [what is the pattern seen in nature?]

From Line 188 Overall, the results [what results??] of this study corroborate the notion that mutational effects tend to be 189 generally but not universally conserved between related species. These results also highlight that the 190 DME on its own may sometimes offer very little predictive or explanatory power when considering 191 cases of apparent constraint, suggesting that evolutionary potential cannot be reliably inferred from 192 measures of the DME. [It would be more organized to move result to the appropriate section and discuss them further in discussion section] [there seem to be a lot of misplacements] consider this in order to avoid confusion and enhance understanding of this present study.

Line 200 [which strain? Please mention which strain for clarity and to avoid confusion]

202-204 [briefly state what the primers chosen and designed were used for]

In the Results section line 488-500 talks about methodology and in part a repetition. Do consider moving to the materials and methods section. Avoid repetition throughout the work which made the work quite lengthy.

General response:

We greatly appreciate the many constructive and thoughtful comments brought to us by the three reviewers and the editor. We have responded to each comment point by point, and these are summarized in the following section. Overall, the most consistent theme of the comments had to do with the misplacement of material that belongs in other sections, repetition, and the overall length. We have made many changes throughout to address these more general comments. The new manuscript has been shortened and contains many changes that were suggested by the reviewers. We feel these alterations have improved the quality of the manuscript and have made it easier for the reader to digest. We have included line numbers for some changes each referencing the “clean” version of the manuscript.

Minor points and formatting:

We found a handful of typos (misspellings) in the supplementary tables. These have been fixed. The main and supplementary figures had their upper-case labels changed to lowercase as specified in the guidelines. Several references have been removed and are no longer cited due to the changes we made to shorten the text as well. A citation of Holt and Gaines (1992) was added in responding to one of the comments.

Point by point response:

REVIEWER #1

The questions asked have been satisfactorily answered to the extent of their research and the methods were quite adequate, robust, and detailed for replicability. However, I have raised just a few issues bordered on clarity and structure.

In this study, we UV mutagenized copper and sulfite sensitive strains of 177 *S. cerevisiae* and *S. paradoxus* and directly recovered mutants from plates containing various 178 concentrations of these chemicals to assay mutational target size [Line 176 - 178 , could you consider rephrasing the last part of the sentence for clarity].

We have rephrased this sentence to be more clear about the process we used to isolate mutants (line 182):

*“In this study, we UV mutagenized copper and sulfite sensitive strains of *S. cerevisiae* and *S. paradoxus*. We then assayed mutational target size by measuring the frequency of resistance mutations when mutagenized cells were plated on various concentrations of these chemicals.”*

Line 179... mutational effect size and pleiotropic cost. [Little explanation of the term may help readers have a better understanding].

We have added some explanation defining these terms (line 184):

“We also subjected thousands of these mutants to a high-throughput phenotyping assay to measure the magnitude of resistance (mutational effect size) and any pleiotropic fitness costs in the absence of stress.”

Line 182-192 seems like a portion of methods, [do consider moving to the materials and methods section to maintain structure and reasonable length of the introductory section].

We agree that this paragraph could be shortened in order to keep the introduction at a more reasonable length. This paragraph has been trimmed to just have a brief explanation of what was done in the study and a one sentence prelude to the results (line 182).

Line 183do not align with the pattern seen in nature. [what is the pattern seen in nature?]

We agree this was confusing. The new version of the paragraph does not make mention of the pattern seen in nature anymore (line 182).

From Line 188 Overall, the results [what results??] of this study corroborate the notion that mutational effects tend to be 189 generally but not universally conserved between related species. These results also highlight that the 190 DME on its own may sometimes offer very little predictive or explanatory power when considering 191 cases of apparent constraint, suggesting that evolutionary potential cannot be reliably inferred from 192 measures of the DME. [It would be more organized to move result to the appropriate section and discuss them further in discussion section] [there seem to be a lot of misplacements] consider this in order to avoid confusion and enhance understanding of this present study.

The reviewer's point about organization is well taken. This paragraph has been modified to introduce how we tested the ideas presented in the rest of the intro rather than providing a results summary (line 182).

Line 200 [which strain? Please mention which strain for clarity and to avoid confusion]

The strains have been specified in the new version as the four strains mentioned in the sentences above (line 197).

“We knocked out the HO gene in monosporic derivatives of each of these four strains by transforming....”

202-204 [briefly state what the primers chosen and designed were used for]

We have clarified this by adding a few words to this sentence (line 200).

*“We used primers JM7 and JM8 from Goldstein and McCusker (1999) for construction of the *S. cerevisiae* HO deletion cassette. For *S. paradoxus* we designed analogous primers with homology to the *S. paradoxus* HO locus (Table S2),”*

In the Results section line 488-500 talks about methodology and in part a repetition. Do consider moving to the materials and methods section. Avoid repetition throughout the work which made the work quite lengthy.

The sentences in these lines (starting at 481) have been simplified and shortened. We have also taken measures throughout to avoid repetition and shorten the manuscript.

REVIEWER #2

The premise of the study is that copper and sulfite resistance evolved in *S. cerevisiae* for winemaking, however, *S. paradoxus* in vineyards has not adapted in these conditions despite occupying the same environment and co-occurring with the sister species. The authors posit different hypotheses to explain the limits of adaptation found in *S. paradoxus* compared to *S. cerevisiae* and measure several factors in resistance mutations: rate, effect size, pleiotropic cost. These hypotheses address interesting questions about the limits of adaptation (e.g. greater adaptive capacity of *cerevisiae*? is *paradoxus* genetically constrained?) and use sister species to test these hypotheses. When they evolved the strains, however, they found similar mutations in both species assayed, suggesting that mutational effects to not explain this difference found in nature.

This makes me wonder if the copper sulfate addition in the winemaking process perhaps has more of an impact than previously thought? I understand that this has been looked into, but could there be some factors that have not been taken into account?

There could certainly be aspects of copper use in winemaking that have underappreciated impacts on microbes. As mentioned in the manuscript, copper sulfate is used in the winemaking process in two ways: (1) As a fungicidal spray on grape vines and (2) in the process of “fining” which removes sulfidic off flavors from the final product. The literature that exists on vineyard microbial ecology points to pretty extreme changes in vineyard soils compared to nearby soils, presumably at least partially due to copper spraying. Copper fining likely imposes little selective pressure on the yeasts because it is added to a final filtered product. That said, there is putatively copper exposure in the must stage due to copper present on the surface of the berries.

There are many ecological unknowns that could play into the species differences we see. For instance, the species may differ in their relative distribution on berries and in soil, yielding differences in copper exposure and relative abundance in the initial must stages. While interesting, the potential ecological explanations for a difference in selective pressures are many and varied. We thought it best to focus on what is known about the two species and confine the discussion mostly to evolutionary explanations because extensive discussion of ecological factors in the winemaking environment would entail lots of speculation. To acknowledge these potential factors, we have made edits to the abstract and the discussion (lines 34-36 and lines 929-932)

I think this is a notable contribution to the understanding of adaptation to toxic environments. I think the experiment is sound and interesting. The text needs clarification and editing throughout to focus the important information, plus organising the information so that it's easily accessible. At the moment there seems to be parallel stories happening and a more clear and explicit delineation of each one would help the reader follow them. I think the writing needs a lot of rearranging. I tried to give pointers where it seemed more important. Here are some examples:

>The Introduction should more organically connect the theoretical predictions for adaptive constraints and DMEs with the background of the history and genetics of copper and sulfite adaptations and tolerance.

We agree the transition is somewhat abrupt. We have made several simplifying edits in this portion of the manuscript (lines 108-118) and have attempted to smooth this transition.

>Perhaps include a bit more information about the statistical methods for each of the hypotheses and follow-up tests (e.g. spot dilution tests). E.g. L266-269 among strains at three concentrations is 1 paired t-test? Was that multiple tests?

We agree the manuscript could benefit from some clarification about the statistical methods. For the paired t-tests we have opted to change our approach for copper in particular after reconsideration given a comment from reviewer #3. For canavanine, the paired t-tests remain because assumptions are not violated. These tests are now explained in more detail as one test that is paired by concentration (line 260).

“To assay differences among strains, we compared our canavanine mutation rates across the three concentrations between species using with a t-test with mutation rates paired by concentration.”

As for the spot dilutions, these are qualitative, and no statistics were used directly. This is mentioned on lines 460, 467.

>Focal strain set for this study included 12 strains derived from the four wild isolates. 150 mutants derived from those strains were sequenced -> total found in the methods, however, it would be helpful in other parts of the manuscript

We agree this is important to mention throughout. The sample size is mentioned in the abstract (line 39), introduction (line 187), methods (line 217), and results (line 602). We also added another reiteration in the discussion (line 810).

>L491-496 are the detailed hypotheses from the introduction. Perhaps swap the formats between the intro and this? Quick reminder in the results and more detailed set-up in the introduction?

We agree that this is a good idea to simplify this in the results and have done so (line 481). In hindsight, characterizing the hypotheses explicitly as “the target size hypothesis” etc. in this way probably did not add anything or aid in understanding. The introduction has two mentions concerning target size, effect size and costs. In the interest of trying to remain concise, we have altered and shortened the wording in the results.

>In Table S3 please indicate columns of the superscript notation. e.g. 2 is column Q, 3 is column H. It wasn't easy to find as they are not unique enough to be searchable

We agree this is a helpful change. This has been implemented in the table.

>L239-243 these lines seem to a better fit with the following section where mutation rate measurements are explained.

We agree that these lines seem out of place looking at them again. We have simply deleted these sentences and separated discussion of canavanine platings and copper/sulfite platings.

>L497 It might be helpful to start/indicate a first subsection addressing the first hypothesis. Leave the short introductory paragraph with the summary and hypotheses.

We agree this is a good idea. This has been implemented with the “Mutant screen” heading following the (shortened) summary paragraph (line 481).

>Check for gene nomenclature throughout. E.g. L723 names of genes should be fixed.

Thank you for the reminder to make sure on nomenclature. In going back through, we found 4 cases of gene nomenclature that either needed to be changed or required more precise wording and made these changes.

However, the instance referenced in this comment we believe is correct (line 726). We are referring to the protein product of *REG1* interacting with the protein product of the *GLC7* gene and its associated proteins.

>I found "The REG1 deletion phenotype is species dependent" especially illuminating and interesting, although I had to read the section multiple times to understand how the experiment was set up to test this interaction. The relationship between REG1 and PMA1 seems very important in understanding the importance of genetic background. Perhaps edit to clarify the importance of this potential interaction?

We agree that the relationship between *REG1* and *PMA1* is very important for understanding the genetic basis of copper resistance. We have revised this section (line 710) to clarify the experiment and interpretation. Specifically, we have emphasized that the potential interaction appears to be intact in both species based on the low pH and copper phenotypes. However, the species do exhibit different phenotypes with respect to growth in the absence of stress, which is a difference that lacks a specific genetic explanation given the data. There is the potential that the greater costs in *S. paradoxus PMA1* mutants are related to the greater costs of *REG1* deletion via the interaction of these two genes and we allude to this on line 896.

In this section (beginning line 710), we reordered the explanation of the experiments for clarity and also changed the abc ordering of the figure panels to reflect that change.

We are hesitant to add any additional statements about the role of genetic background in this interaction. We think the key difference between species being highlighted is the growth defect difference, not the potential gene interaction.

>Figures: boxplots might work best as violin plots.

We had the same thought in preparing the manuscript. For the costs plot of all the permissive phenotypes in particular, it would be nice to get a more complete sense of the distributions via a violin plot. However, given the spatial constraints and the large number of categories, we felt the violin versions of these plots were very hard to interpret at a glance. The boxplots in these figures we feel convey the critical information of the presence or absence of costs, and show the conditions in which the species differ.

As for the sequenced mutants, the lower sample sizes allow for all the points to be present on the plot. A violin we feel would likely have a similar effect of making the plots more difficult to read in the constrained space and would not add any information critical to interpretation.

>The discussion should be re-written and shortened to follow more explicitly with the title of the subsections the different stories the paper is addressing. Additionally, some of the discussion could be included in the results (e.g. L1000-1003)

We agree there is some unnecessary repetition and crossover between sections. Improved delineation would be beneficial to the clarity of the manuscript and allow us to shorten it without losing any substance. We have tried to eliminate redundancies and to keep the sections more delineated. That said, there are instances where crossover between sections we feel is beneficial. For instance, when we discuss the costs results, confining the discussion purely to phenotype without mentioning the sequencing data, leaves the interpretation of the phenotypes incomplete.

As for the mention of mutation rates on line 1000 (now line 972-974), although it is a restatement of a result, we feel this is not a statement that belongs in the results due to its intent of providing context to the following speculative statement about the rate of de novo *CUP1* duplications, which is an unknown. We think this statement requires the context of the number of cells screened. Without this reiteration, the subsequent speculation would likely require the reader to go back through the methods and tables to get the number of cells screened so that they could figure out where the 10^{-8} number is coming from.

REVIEWER #3

I enjoyed thinking about the experiments in the manuscript "The distribution of beneficial mutational effects between two sister yeast species poorly explains natural outcomes of vineyard adaptation." The paper seeks to explain the observation that *Saccharomyces cerevisiae* has readily adapted to copper and sulfite in the context of vineyards, while the sympatric and closely related species *Saccharomyces paradoxus* has not. They acquired mutant haploid and diploid colonies on 8-12 levels of copper and sulfite from UV mutagenized

and mock pools derived from two isolates from each species isolated off trees in Europe. They phenotypically screened nearly 4000 isolates and whole genome sequenced 150 haploids. They tested whether there are species-level differences in mutation rate, beneficial mutation effect sizes, or the pleiotropic costs of beneficial mutations in nonstressful conditions. Although some minor differences were reported, overall, they concluded that none of these factors could explain the observed adaptive outcome seen in vineyards.

My comments and questions are generally made to better understand how certain experimental decisions were made and whether analysis choices could have influenced the interpretation.

1. I had a hard time following how mutants were picked in the methods section, and some information seems to be missing. I suspect a flow diagram might help to see how the provided numbers are linked to genotype and drug concentration they were isolated from; from the subset of 3024 haploids and 720 diploids that produce colonies, what ended up being sequenced? Figure S1A could be expanded to include numbers and more specifics to show how you go from two species x two backgrounds x 8-12 concentrations to the number of mutants selected for phenotyping and then sequencing.

We agree that additional clarity could be provided. We have merged the recovery concentrations of each of the sequenced mutants into table S3. We have also emphasized that selection for sequencing was independent of recovery concentration and was only influenced by resistance (line 366).

“Both spontaneous and induced mutants from a wide variety of recovery concentrations were sequenced (Table S3), but selection for sequencing was done independent of this variable and only considering ΔAUC .”

We have also emphasized that recovery concentrations were recorded when mutants were isolated and are present within the phenotyping datasets (line 308).

“For each mutant, we recorded recovery concentration and three aspects of colony morphology....”

After experimenting with a few ideas, we think that a flow diagram that includes all the mutant picking information would become quite busy given the two species x two ploidy states x two stressors structure of the data and would be redundant with information present in tables S3 and S4. A simpler version that collapses stressors/species/ploidy states to fewer numbers would also not aid in understanding or be informative in our view. However, we think adding some of this information to figure S1a is a helpful suggestion that makes our workflow easier to understand. In the updated version, we have added mention of the numbers of cells screened, numbers of isolates phenotyped, and the number of strains sequenced. The caption has also been edited slightly to accommodate this additional information.

2. I am curious whether mutants acquired at higher drug concentrations differ in mutation type or effect size. The text states that mutants were recovered from higher concentrations in *S cerevisiae* than from *S paradoxus*, but I didn't see anywhere else in the manuscript where this was directly interrogated. For the copper sequenced isolates, it was stated that they were evenly distributed across the four haploid ancestral strains - what was the split among drug concentration of isolation? There are a number of recent papers in *Candida albicans* showing that the mutations acquired at lower drug concentrations are different genotypically and phenotypically than the mutations at higher concentrations, and it would be interesting to see whether that is also true here.

This is a very interesting set of points. Recovery concentration and resistance among the mutants' phenotype did not show a significant relationship among sequenced mutants ($p = 0.08$). There was also no significant relationship among the diploid copper mutants ($p = 0.26$). However, there was a weak though significant positive relationship in haploid copper mutants (Pearson's $r = 0.2$, $p < 10^{-5}$). For sulfite, there was no relationship between recovery concentration and delta AUC in haploids ($p = 0.16$) or diploids ($p = 0.25$). We have now made mention of these results in the text (lines 550 and 557).

"Recovery concentration was weakly though positively correlated with ΔAUC in haploid copper mutants (Pearson's $r = 0.2$, $p < 10^{-5}$), but not diploid copper mutants ($p = 0.26$)."

"Recovery concentration did not correlate with ΔAUC in haploid or diploid sulfite mutants ($p > 0.15$ in both cases)."

As for genetic identity of mutants, all the mutants classes identified as causal are dispersed across many recovery concentrations and this can be seen in table S3 now that the merging of recovery concentration has been performed. This should have been included at the outset in hindsight and we thank the reviewer for pointing this missing piece of information out. The recovery concentration of all the mutants is available in the datasets on figshare as well for all of the other mutants that were not sequenced and this has been mentioned in the methods as stated in the response to the prior comment.

Overall, we agree that there can be very interesting differences among mutations and how their effects interact with concentration. However, there were not strong patterns that we were able to discern that suggested recovery concentration was playing a large role in determining genetic identity or resistance of the mutations we found.

3. For testing mutational target site, I wonder if collapsing all of the analysis to a paired t-test within drug, without accounting for the other variables (strain, concentration) has obscured some interesting patterns. Given that the underlying data is count data, a t-test is probably not the most accurate test, regardless. Based on Figure 1, it looks like *S cerevisiae* has a higher mutation rate than *S paradoxus*, and given that this result is central to the conclusions, I think it's really critical to ensure the statistics are done correctly.

We agree that the stats here need to be given careful consideration because the inference of equal versus nonequal mutation rates has implications for our overall results. We have given a lot

of thought to the stats on this figure in particular. In this test, we were taking the mutation rate measurements for both species and pairing them by concentration. The null hypothesis is that these pairs of rates show a difference of zero. We found that when analyzed this way, mutation rates were not found to differ between species for canavanine or copper. Upon reconsideration, the paired t-test does have one key issue based on the assumptions of the test for the copper comparisons. A paired t-test assumes that the differences between samples are normally distributed, and this assumption was inconsistently met across the tests for copper. For canavanine, this assumption is met, and we think the test is appropriate.

Given this complication for copper, we have changed the approach to be count based using a chi-square framework. In doing this, we make the tests between copper and sulfite more similar. That said, the chi square framework entails that each concentration be analyzed separately. After performing these analyses, we found that at several of the lower concentrations there are significant species differences favoring a greater mutation rate in *S. cerevisiae*, as could already be seen in the figure. However, at the higher concentrations these differences disappear as the mutation rates get smaller. In the updated figure we have signified individual concentrations where both *S. cerevisiae* ancestors have higher mutation rates than *both* *S. paradoxus* strains. This conservative approach we think is warranted because if the overall pattern is driven by only one of the ancestral strains, then we do not want to call that a species difference. We have also more directly addressed these apparent differences in the text. Despite the stats, differences at the lowest concentrations are potentially the most likely to be confounded for two reasons: 1) Physiological escape is more likely as the concentration decreases, meaning a larger fraction of the colonies are not true mutants and 2) The small differences in basal resistance among the ancestral strains will likely bias mutation rates the most at the lower concentrations due to the smaller relative differential in resistance needed to be gained compared to the higher concentrations. We have added these caveats to the results.

We think that this alternative approach and presentation is more complete, nuanced, and avoids potentially misleading aspects of the prior analysis. The message we think these data convey is that there are differences at the low concentrations, but these differences may not reflect a target size difference due to confounding effects at low concentrations.

In light of these changes, we think the other chi-square tests in the previous version of the manuscript looking at the distributions of mutants across concentrations are confusing and add little. These have been removed.

4. The phenotypic assessment is dependent on the Δ AUC calculation, and I wonder whether some aspect of the analysis could be averaging out some of the variation among mutation effects.

- With respect to ancestral values - Were the ancestral AUC values variable? How many replicates of the ancestral strains were measured to ensure precision in the ancestral average AUC? Was the normalization procedure needed?

There were subtle differences among ancestral strains. These numbers can be derived from the figshare datasets or calculated from information (AUC - Δ AUC) in table s3. In the haploids there were two technical replicates of the ancestors and for the diploids there were eight (lines 292 and

301 (4 replicates on the 96 plate x 2 technical replicates when plate is collapsed)). These values, though slightly variable, are highly precise. For example, the greatest deviation between technical replicates for haploid copper AUCs in ancestors we saw was 0.0032 and the most variable set of measurements for ancestral AUCs for diploid copper were:

0.01589661
0.01585237
0.01656553
0.01475000
0.01497243
0.01470030
0.01556115
0.01750960

These numbers signify reduced colony size on 0.02mM and essentially no growth on any concentration beyond this. Since most copper AUCs were far larger than these numbers, the ancestral variability was small compared to the variability among mutants.

Sulfite ancestral values are discussed further in the response to the comment about defining escapees. Overall, the normalization to ancestral values has a miniscule effect of the measurements for copper and a slightly greater effect for sulfite due to the magnitude of the differences among ancestors. We think the normalization to the ancestral AUCs makes the comparisons among species as fair as possible and accounts for the basal differences.

It doesn't seem obvious from a visual assessment of Figure S2 (but the effect might be subtle). If you look at the technical replicates that were at different places in the plates, is the data consistent with an edge effect?

Permissive phenotypes are heavily influenced by edge effects. With the size of the colonies on the outer layers being consistently larger than those on the inner layers. For copper resistance measurements, edge effects are massively reduced and the highest concentration at which a clone produces a colony becomes the driver of its AUC rather than its position. This is presumably for two reasons. First, irrespective of plate position, a sensitive strain will disappear at higher concentrations and a resistant strain will remain. Second, as the concentration increases, more colonies disappear from the plate, reducing the average number of neighboring competitors per resistant clone.

Technical replicates that were included for the ancestors produced highly consistent AUC values as mentioned in the response to the prior comment. These values are not consistent with an edge effect.

It is very difficult to determine if technical replicates for mutants are consistent with an edge effect because they were only performed for the diploids and they are always in adjacent positions based on how we collapsed the plates to 1536 format. Thus, they only ever differ by one layer at most.

For haploid copper mutants, there is a weak though significant positive correlation between layer and AUC. This means that on average the strains on the interior of the plate have greater AUCs. Inspecting images from this experiment shows that this is indeed the case. However, we do not interpret this not as a result of an edge effect because the images clearly show more colonies in the center of the plate compared to the edges at higher concentrations. Edge effects may affect colony size, but they should not affect presence or absence of a colony. As such, this relationship appears to be driven by it actually being true that more resistant strains are biased for the center of the plate due to our sampling scheme rather than anything intrinsic to the center of the plate.

For these reasons, we do not think edge effects were a confounder for resistance measurements.

- How well does Δ AUC correlate with growth at one of the higher levels, such as 0.3 mM?

They correlate quite well. Growth at a higher concentration is intrinsically linked to AUC. Although similar, the AUC measurement contains more information about those strains that are less resistant and fail to grow at the higher concentrations.

As an example, Pearson's r for AUC and growth at 0.3mM in the haploid dataset is 0.57 . ($p < 2.2e-16$). Only the most resistant strains are producing colonies and all the less resistant strains have colony sizes of zero. This concentration is thus a good proxy for high resistance but it essentially bins all of the less resistant strains into a single category as not growing at all.

- I was confused by the statement that only mutations that were 3SD above the ancestor were included in downstream analysis. In Figures 2, 5, and S5 it looks like there are many mutations very close to 0 Δ AUC. As above, were the mutants within 3SD (termed by the authors as physiological escapees, but this could also be low s mutations?) most likely to come from low drug plates?

The standard deviation of the ancestors were quite tight for copper (for example the largest sd for haploid copper was 0.00228, the lowest was 9.9×10^{-5}). This explains the values close to zero retained in figures 2 and S5. In figure 5, no such exclusion was carried out for sequenced mutants.

We think this is justified because the results in figures 2 and S5 require the removal of escapees for proper interpretation, whereas the results in figure 5 for sequenced mutants do not.

Some of the low AUCs we excluded could be very low s mutations. However, at those levels for copper they are essentially phenocopying the ancestors, and we do not think their inclusion as mutants with a detectable copper phenotype is prudent.

This 3sd measure was most critical for sulfite, where the ancestral spread was far larger (0.365 for the most variable ancestor). This lowered our confidence that the observed phenotypes were due to a mutation rather than something like locally lower sulfite concentrations. We did not want to include low confidence measurements in the results or low confidence mutants in the sequencing. As such we used the 3sd criteria as a filter. This may have missed some true mutants, but as stated in the text, the 38 sulfite mutants we analyzed were the only ones based on our criteria that could be ruled out as physiological escapees.

- If you look at the higher effect mutants, does anything jump out about where they came from?

The largest effect mutants for copper have varying recovery concentrations spanning nearly the entire range, and there is no clear pattern such as the largest effect mutants come from intermediate concentrations. As for the genetic identity of mutations, *PMA1* mutations had the largest effect size, but there was a high variance in recovery concentration associated with mutations in this gene.

5. Given that the paper's stated goal was to understand the observation that *S cerevisiae* adapts more readily to copper and sulfite than *S paradoxus* in nature, I had expected the diploids to be featured more prominently in the manuscript than haploids. I was very surprised to see that only the haploid copper mutants were sequenced, and I'm wondering why that decision was made. I do not suggest additional sequencing lightly, but it would be very interesting from a ploidy evolution perspective, and I think more ecologically relevant, to also look at the mutations that arose in the diploid strains.

This study was originally construed as a screen of haploid strains only. This choice was rooted in the idea that both dominant and recessive mutations could be analyzed, and that although Haldane's sieve can act in yeasts, there also exist many ways for diploids to subvert Haldane's sieve via loss of heterozygosity.

Upon obtaining phenotypes for haploid mutants and selecting mutants to sequence, it was pointed out that if recessive mutations were driving the species difference we saw for copper, the answer may be different in diploids. As such, we conducted our diploid screen subsequently post sequencing. We agree the data would have been well complemented by diploid sequenced clones, but the decisions made prior precluded this. That said, any large effect dominant mutation should be detectable in the haploids. The diploid mutants obtained in this screen were also not stocked. As such, adding diploid sequencing data would entail another screen of our mutagenized pools.

We have added a brief discussion of the potential biases introduced by only sequencing haploids to the discussion (line 920):

“Overall, the choice to sequence only haploids may have introduced biases for loss of function mutations with some recessive character among our sequenced mutants given the causal mutations in our dataset. These mutations may be of limited ecological relevance due to most natural isolates being diploid. That said, any large effect dominant mutation would also be accessible in haploids, and PMA1 mutations were shown to have an effect of copper resistance in diploids. However, sequencing diploid copper and sulfite mutants would likely offer additional insight into the genetics and evolution of these two traits”

6. Can you be more specific for the "unknown" mutations? Were there multiple well-supported variants in these lines?

We have clarified this in the text. (line 675)

“We also performed these comparisons for the subset of strains that lacked any of the above mutations that were assigned as causal. These strains were subsequently designated as “unknown causal variant” in these analyses.”

The only well supported variants were those in *PMA1*, *REG1*, *KSP1*, *RTS1*, CHRVIII aneuploidy and CHRIII aneuploidy. All other variants including singletons are included in table S3 and some may very well be causal. However, without experimental validation, we lack the power to assign any of these mutations as causal.

7. The discussion talks about how the known mechanisms for copper and sulphite resistance in nature were largely not obtained through their screen. Is this due to the biases of UV-C mutagenesis, not just that these mutations are rare and hence 'beyond the detection limit of normal screens' (L972)? A discussion of the mutants that arose through mutagenesis vs. the mock would also be helpful (if some of the mock mutants were indeed sequenced).

Some mock mutagenized strains were sequenced, and this is noted in Table S3. We have also now made note of this in the text (line 366).

“Both spontaneous and induced mutants from a wide variety of recovery concentrations were sequenced (Table S3), but selection for sequencing was done independent of this variable and only considering ΔAUC ”

Most of these had mutations similar to the UV mutagenized sequenced mutants with *REG1*, *PMA1*, CHRIII, and CHVIII mutants among the spontaneous mutants (ex. YJF4413 and YJF4514). We think drawing attention to this comparison, since mutagenized and mock mutagenized mutants are by and large similar, is extraneous for the main points of the paper. However, the information is available in table S3.

As far as the “detection limit”, we agree this could be confusing because we are not talking about UVC induced mutations, and that was unclear. We presume that these mutations would almost certainly be spontaneous if present, and that these mutations would be easily detectable due to their extremely large phenotypic effect. As such they are indeed likely beyond the detection limit of the screen but this limit refers to the spontaneous rate limit rather than the induced rate limit. We have clarified what we mean by beyond the detection limit of the screen (line 972).

We think this is an important point to make because in our experience with this project (and others), this empirical insight of extreme rarity of *de novo CUP1* duplication has proven astonishingly true. We agree that the potential bias of mutagenesis should be addressed in the text, and we make mention of it several times. However, we also contend that our lone strain with a *de novo CUP1* duplication is likely not an artifact of our chosen method, but rather (at least in part) a reflection of the low rate at which these mutations occur.

We have made several edits to this portion of the manuscript (beginning line 971) and it now reads

*“We screened $\sim 10^8$ mutagenized cells in total and recovered a single case of *de novo CUP1* duplication. Given these mutations are unlikely to be induced via UV exposure, the mutation rate for *CUP1* amplification may be on the order of 10^{-8} , but with a sample size of only one the rate could be*

even lower. Among the sulfite mutants we sequenced, we did not observe any rearrangements involving SSU1, meaning the rate of (spontaneous) occurrence of these mutations is lower than the detection limit of the screen we performed. Translocations and inversions are relatively rare compared to other types of mutations, and it is conspicuous that these rare classes of mutants have repeatedly been selected for in sulfite exposed strains."

8. Both the Introduction (over 6 pages) and Discussion (over 11 pages) are quite long and could be tightened up to focus on the things that matter most to the reader and to place the results in the context of the literature. There are essentially no references for the first several pages of the discussion, by example. Other parts of the discussion verge quite deep into the mechanistic basis of different types of mutations, which, although interesting to read about, were fairly far afield of the experiments that were done.

We have revised the introduction and discussion to be shorter and we agree that little to none of the important points of the paper have been lost via these changes. Mostly we made simplifying edits and tried to reduce redundancy. However, for the mechanistic basis of the different mutations on phenotype, we contend that aside from the small alterations we have done that these should remain intact. For *CUP1* duplication in particular, this one strain we found has strong implications about the natural pattern and raises many questions about different rates of different duplication mechanisms across species. As for the other genes, several of the mechanistic insights are novel and connect to prior results in important ways. In keeping with trying to make the work shorter, there have been some changes in the mechanistic discussions, but the bulk of these have been kept in the updated manuscript.

ASSOCIATE EDITOR COMMENTS

In addition to the suggestions above, I found myself puzzled when reading the abstract that it focused so much on evolutionary constraints without mentioning ecological or phenotypic constraints. It could be, for example, that niche differences between *S. cerevisiae* and *S. paradoxus* make *S. cerevisiae* differentially predisposed to domestication (e.g., tendency to colonize vats of wine, pre-existing copper and sulfite tolerance, artificial selection favouring *S. cerevisiae*, e.g., because of the flavour of the wine). There are many reasons that one species may have repeatedly adapted to wine brewing in the presence of copper and sulfite, which do not relate to evolutionary adaptability.

The main text didn't raise this concern so much in my mind because it focused on a narrower question: do they have differential access to mutations? But the abstract seemed misleading, I believe, in claiming that the explanation for the preponderance of *S. cerevisiae* in copper/sulfite-adapted wine yeast is due to their greater ability to adapt: "...*S. paradoxus*, has not adapted to these chemicals despite being consistently present in sympatry with *S. cerevisiae* in vineyards. This contrast represents a case of apparent evolutionary constraints favoring greater adaptive capacity in *S. cerevisiae*."

Put another way, population size matters enormously for evolutionary adaptability, even if the target size in the genome and the distribution of fitness effects are the same. Thus, ecological differences could allow *S. cerevisiae* to adapt to conditions that *S. paradoxus* cannot, simply because they survive in high enough numbers to adapt.

The paper might benefit from a short discussion of the role of ecological differences and a rewrite of the abstract to better reflect that it is a null hypothesis to be tested that *S. cerevisiae* & *S. paradoxus* are equally able to adapt to copper/sulfite.

These are all very good points, and we agree that the abstract should be altered so as not to mislead. The text has been altered to read (line 33):

“S. paradoxus, has not adapted to these chemicals despite being consistently present in sympatry with S. cerevisiae in vineyards. This contrast could be driven by a number of factors including niche differences or differential access to resistance mutations between species.”

To format for Genetics and get the abstract down to 250 words, we also made a few simplifying edits throughout the abstract.

We feel this adequately addresses the possibility of something other than mutational constraints leading to the differences seen in nature and reflects that we are focusing on the possibility of differential mutational access.

Ecological factors such as population size and source/sink dynamics may indeed be playing a large role and we have added a few sentences to the discussion to address this possibility with an added citation to Holt and Gaines (1992), which discusses in detail how source/sink dynamics can impact the fate of mutations adaptive outside the fundamental niche (line 929):

“The absence of these large effect mutations in S. paradoxus could be driven by many factors including differences in population size in vineyards, niche differences such as contrasting in source/sink dynamics in vineyards, or differential mutational access (Holt and Gaines 1992).”

August 24, 2024

RE: GENETICS-2024-307376

Dear Dr. Longan:

I am pleased to accept your manuscript entitled "The distribution of beneficial mutational effects between two sister yeast species poorly explains natural outcomes of vineyard adaptation" for publication in GENETICS, pending minor revision.

Please submit your revision along with a brief description of how you modified the manuscript in response to the handling editors' concerns and suggestions, which can be viewed at the bottom of this email. Most important are a few potential inconsistencies and instances of repetition. When revising the ms., please make an effort to shorten it, because that almost always improves a manuscript. We urge authors to heed the advice of Strunk and White: "omit needless words"¹. I expect that you should be able to submit a revised manuscript within 30 days. A suitably revised manuscript will be acceptable for publication; I don't expect to send it out for review.

Follow this link to submit the revised manuscript: Link Not Available

Thank you for submitting this story to Genetics.

Sincerely,

Sarah Otto
Associate Editor
GENETICS

Approved by:
David Begun
Senior Editor
GENETICS

Reviewer comments:

Not sent out for re-review.

Associate Editor comments:

This paper asks whether closely related species differ enough in mutational constraints to explain the ability of only some species to adapt to a novel environment. Focusing on the sister yeast species *Saccharomyces cerevisiae* and *paradoxus* exposed to copper and sulfite, the authors obtain a series of mutagenesis lines and demonstrate that there is little difference in the number and effect size of potential adaptive mutations. These results point to other factors (including rare structural changes and ecological differences) to explain why *S. cerevisiae* and not *paradoxus* has adapted to viticulture conditions with high exposure to copper and sulfite. I believe that this manuscript will be of broad interest to the audience of Genetics and appreciate the changes that the authors have made in light of the reviews. I recommend accepting the manuscript, but I offer some additional minor suggestions below to address before submitting a final revision.

Line 313 - It was unclear to me how long the master 1536 plates were grown before replica plating. Does the size of the colony on the master plate influence the size on the copper or sulfite plates? I wondered whether this could bias the phenotyping. Comment briefly?

Line 361 - To clarify why you sequenced haploids, change to (?): "For copper mutants, we selected 120 mutants for sequencing, evenly distributed across our four haploid ancestral strains. We focused genomic analyses on haploids to avoid confounding effects of dominance and to facilitate the identification of mutations (e.g., eliminating many false positive heterozygous calls)."

Line 388 - 91,840 what? potential SNPs?

Line 507 conflicts with the statement on line 510. Meaning?

Line 512 and surrounding - In the paragraph on sulfites and later on line 537 for copper, you explicitly exclude physiological escapees using the protocol mentioned in the methods (within three standard deviations of the ancestral measurements). Why not perform this protocol here first when considering the mutational target size in copper?

Line 540 - I could not make the numbers match up. Here 75.6% of copper data are said to be retained after excluding physiological escapees, but line 539 says that there were 1109+309 retained mutants out of 8704 obtained in copper (line 506), which is much less than 75.6%. I might have misunderstood, but I likely won't be the only one to do so.

Line 562 - Change to (for clarity?): "difference, haploid *S. paradoxus* exhibiting larger-effect mutations in copper, does not align"

Line 797 - Paragraph could be shortened (seemed speculative and inconclusive).

Line 968 - abundance -> in abundance

There remain several sections that are repetitive across the paper, making the paper longer than it needs to be. Where I noticed this was:

* Lines 496-501 repeats results presented in the methods. Drop down to the last sentence only?

* Lines 535-540 repeats results that are either presented (for sulfites on line 519) or should be presented (for copper around line 506) earlier. The discussion of dropping physiological escapees should thus be made once in the "Mutational target size" section and only alluded to in the "Mutant effect size" section.

* Lines 524-527 seemed repetitive and could be deleted. (Besides, the results cannot show "equal" mutagenesis, only a lack of significant differences.)

* Lines 608 and 621 (plus some surrounding material) are repetitive.

* The most extensive repetition concerns the interpretation of genetic changes, which is discussed at length in both the results and the discussion. I would suggest trimming down one of these. In particular, if a lot of the genetic material was moved from the Discussion and merged (cutting repetition) into the Results, that would streamline the Discussion, which seems long (nine pages) relative to the new perspectives that it provides. Besides, learning some things (e.g., about PMA1 and CUP1) in the Results and some in the Discussion makes it harder for the reader to process the material.

Reviewer comments:

Not sent out for re-review.

Associate Editor comments:

This paper asks whether closely related species differ enough in mutational constraints to explain the ability of only some species to adapt to a novel environment. Focusing on the sister yeast species *Saccharomyces cerevisiae* and *paradoxus* exposed to copper and sulfite, the authors obtain a series of mutagenesis lines and demonstrate that there is little difference in the number and effect size of potential adaptive mutations. These results point to other factors (including rare structural changes and ecological differences) to explain why *S. cerevisiae* and not *paradoxus* has adapted to viniculture conditions with high exposure to copper and sulfite. I believe that this manuscript will be of broad interest to the audience of *Genetics* and appreciate the changes that the authors have made in light of the reviews. I recommend accepting the manuscript, but I offer some additional minor suggestions below to address before submitting a final revision.

Line 313 - It was unclear to me how long the master 1536 plates were grown before replica plating. Does the size of the colony on the master plate influence the size on the copper or sulfite plates? I wondered whether this could bias the phenotyping. Comment briefly?

The master plates were derived from the arrays of mutants and were grown for two days. This is now noted in the manuscript. The size of the pin is much smaller (<10x) than the size of a wildtype colony. During a transfer, the robot pin is pressed down onto the plate to achieve consistent adherence of cells to the pin. We previously calculated that after 2 days of growth there are $\sim 10^7$ cells per colony (PMC9560512) and $\sim 500k$ cells are transferred (<5%). If variance in the number of cells adhered to the pin were a substantial confounder, we might expect smaller colonies to lead to the transfer of fewer cells begetting small colonies in all tested conditions. Our costs \sim resistance results seen in Figure S5 would not be expected in this case. We saw that costlier mutants produce relatively larger colonies on higher concentrations of copper than their less costly counterparts.

That said, we have no way of excluding a small effect of master plate colony size on subsequent phenotyping. One other observation that helps alleviate this concern is that edge effects (large colonies on the edges) develop and worsen over the course of multiple days. Thus, on the first day colonies are all very small but quite uniform despite being pinned from a master plate with noticeable edge effects. Also, even if the number of cells transferred has variance associated with it, on copper plates this variance is expected to have no influence on the presence or absence of a colony after three days of growth as long as a position is inoculated. We expect a resistant strain to produce a colony and a sensitive one to fail to produce a colony irrespective of the number of cells transferred so long as that number is nonzero.

Line 361 - To clarify why you sequenced haploids, change to (?): "For copper mutants, we selected 120 mutants for sequencing, evenly distributed across our four haploid ancestral strains. We focused genomic analyses on haploids to avoid confounding effects of dominance and to facilitate the identification of mutations (e.g., eliminating many false positive heterozygous calls)."

We agree this is an improvement. The change has been added.

Line 388 - 91,840 what? potential SNPs?

This number refers to the total number of variants (SNPs and indels) in the vcf when all *S. cerevisiae* strains were combined. This has been noted in the text to reflect what is presented in table S5.

Line 507 conflicts with the statement on line 510. Meaning?

We understand that these sentences are confusing reading them again. To clarify, the sentence on line 507 is a comment about the maximum recovery concentration for single mutants. For example, we saw mutants on the 0.6mM plates for *S. cerevisiae* but not *S. paradoxus* (table S4). The sentence on line 510 is referring to the statistical treatment of those higher concentrations where there is no significant difference. We think the simplest solution here is to remove the first sentence. This information is present in table S4, but it is not critical to the discussion about comparing mutation rates.

Line 512 and surrounding - In the paragraph on sulfites and later on line 537 for copper, you explicitly exclude physiological escapees using the protocol mentioned in the methods (within three standard deviations of the ancestral measurements). Why not perform this protocol here first when considering the mutational target size in copper?

This is a great question. The issue here has to do with the fact that physiological escapees could only be identified in hindsight after phenotyping and that copper and sulfite mutants had such different rates of physiological escape. Because there were 1000s of copper mutants recovered in the plating assay that were not phenotyped, we cannot confidently assign any of these strains as true mutants or escapees. Escapees were far less frequent for copper than sulfite, but without exhaustively testing all recovered colonies the precise frequency is unknown. For sulfite, the situation is different. Almost all of the colonies appeared to be escapees after phenotyping meaning that the number of true mutants that were not phenotyped from the original plating was likely small. Because the 31 strains with substantial sulfite phenotypes were the only ones that we were confident were true mutants, these were the only ones retained for the mutation rate comparisons. For copper many hundreds of mutants that appeared in the plating assay that were not phenotyped were likely true mutants, but this is an unknown not easily accounted for.

Line 540 - I could not make the numbers match up. Here 75.6% of copper data are said to be retained after excluding physiological escapees, but line 539 says that there were 1109+309 retained mutants out of 8704 obtained in copper (line 506), which is much less than 75.6%. I might have misunderstood, but I likely won't be the only one to do so.

This difference is accounted for in a similar manner as the response to the comment above. We have attempted to clarify in the text by adding an additional sentence to the methods emphasizing the numbers mutants that were phenotyped. In short, 8,704 is the number of colonies we observed in the copper plating assay, which is a substantially larger number than the number of mutants we phenotyped (1512 + 360 = 1872). The 75.6 number comes from $(1512 \text{ haploids phenotyped} + 360 \text{ diploids phenotyped}) / (1107 \text{ nonescapee haploids} + 309 \text{ nonescapee diploids}) = 75.6\%$.

By the same token for sulfite, the numbers come from $(1512 \text{ haploids phenotyped} + 360 \text{ diploids phenotyped}) / (31 \text{ nonescapee haploids} + 7 \text{ nonescapee diploids}) = 2.0\%$.

Line 562 - Change to (for clarity?): "difference, haploid *S. paradoxus* exhibiting larger-effect mutations in copper, does not align"

We agree this flows better and is more complete. This change has been made.

Line 797 - Paragraph could be shortened (seemed speculative and inconclusive).

We agree this paragraph is speculative and can be shortened. We think the critical thing to retain is that an ecological inference based on the costs results for copper is not appropriate without a class of mutations that show a cost in one species and not the other. This paragraph has been shortened in the updated version.

Line 968 - abundance -> in abundance

We agree this is better.

There remain several sections that are repetitive across the paper, making the paper longer than it needs to be. Where I noticed this was:

*** Lines 496-501 repeats results presented in the methods. Drop down to the last sentence only?**

We agree this is unnecessarily repetitive. We have removed the repetitive portion and only retained the first sentence referencing the methods and the last sentence stating that our calculations give rise to a reasonable expectation of site level parallelism. We made a couple of slight edits to the last sentence to reference table S4 earlier and to smooth the transition following removal of the prior sentence.

*** Lines 535-540 repeats results that are either presented (for sulfites on line 519) or should be presented (for copper around line 506) earlier. The discussion of dropping physiological escapees should thus be made once in the "Mutational target size" section and only alluded to in the "Mutant effect size" section.**

Physiological escapees can only be identified in hindsight following effect size phenotyping. Because so many of the sulfite “mutants” seen in the plating assay are in fact escapees, we think it is inappropriate to make a mutation rate comparison using the number of colonies recovered. This was not the case for copper, where physiological escapees are not nearly as common. As such, colony count was used to assess mutation rates in this case. Because sampling of copper resistant colonies for phenotyping was not near exhaustive (1872/8704), we cannot make an inference about the true rate of escapees and apply a correction based on the phenotyping data. Because for sulfite the sampling of true mutants was such a rare occurrence, we can more safely infer that nearly every colony recovered in the plating assay was an escapee. By this logic we only retained true mutants identified via phenotyping when making mutation rate comparisons.

Stated another way, physiological escapees could not be removed for mutation rate calculations for copper because there were 1000s of mutants that were not phenotyped and the rate of physiological escape was relatively low. Physiological escapees for sulfite should be removed because nearly all of the colonies were escapees.

We agree that bringing up escapees once would be more streamlined. However, we feel this is required to be both as transparent and accurate as possible. We have made a few edits to try to make this more clear but feel that the treatment of the copper and sulfite differently in this respect warrants mention of the escapee adjustment in both sections.

*** Lines 524-527 seemed repetitive and could be deleted. (Besides, the results cannot show "equal" mutagenesis, only a lack of significant differences.)**

We agree there is no loss of substance with this change.

*** Lines 608 and 621 (plus some surrounding material) are repetitive.**

These two lines are unnecessarily repetitive. We have made changes in this section to avoid repetition.

*** The most extensive repetition concerns the interpretation of genetic changes, which is discussed at length in both the results and the discussion. I would suggest trimming down one of these. In particular, if a lot of the genetic material was moved from the Discussion and merged (cutting repetition) into the Results, that would streamline the Discussion, which**

seems long (nine pages) relative to the new perspectives that it provides. Besides, learning some things (e.g., about PMA1 and CUP1) in the Results and some in the Discussion makes it harder for the reader to process the material.

We agree in reading these sections back to back that they echo each other too much. The approach that we have taken is to trim the discussion as much as possible so as not to repeat results and to also remove extraneous sentences that were over-explaining. We hesitate to transplant things in the discussion into the results because we fear going beyond the level of detail needed to lay out the motivation for the follow up experiments may detract from the message and also seem out of place. The largest deletions came in the *PMA1* and *REG1* sections of the discussion, and we do not think any substance was lost. We also made a few simplifying edits in the section discussing aneuploidy that shortened the discussion further. The details about KSP1 and RTS1's putative mechanisms of action we also think should not be transplanted to the results due to the lack of follow up experiments.

Throughout, we have also made many simplifying edits that shortened the text. The most substantial of which that is not mentioned in any of the comments above is in the results. We drastically simplified our explanation of the pleiotropic costs results, and we feel this did not remove anything essential.

There was also a very small edit made to figure 2, which was simply removing one of a pair of duplicated text boxes that were overlaid.

Overall, we thank the editor for the thoughtful comments. We hope the updated manuscript has adequately addressed the main issues these comments presented, namely the discrepancy between the number of colonies counted in the mutation rate assay, the removal of escapees, and also the repetitive and wordy elements of the discussion.

September 23, 2024
RE: GENETICS-2024-307376R1

Dr. Emery Roger Longan
University of Rochester Department of Biology
Biology
1308 Elmwood Avenue
Apt 4
Rochester, New York 14620

Dear Dr. Longan:

Congratulations! We are delighted to inform you that your manuscript entitled "The distribution of beneficial mutational effects between two sister yeast species poorly explains natural outcomes of vineyard adaptation" is acceptable for publication in GENETICS. Many thanks for submitting your research to the journal.

I particularly appreciated the care taken with the revisions and have no further suggestions.

To Proceed to Production:

1. Format your article according to GENETICS style, as discussed at <https://academic.oup.com/genetics/pages/general-instructions>, and upload your final files at <https://genetics.msubmit.net>.
2. Your manuscript will be published as-is (unedited-as submitted, reviewed, and accepted) at the GENETICS website as an Advanced Access article and deposited into PubMed shortly after receipt of source files and the completed license to publish. Please notify sourcefiles@thegsajournals.org if you do not wish to publish your article via Advanced Access.
3. We invite you to submit an original color figure related to your paper for consideration as cover art. Please email your submission to the editorial office or upload it with your final files. You can submit a small-sized image for evaluation, and if selected, the final image must be a TIFF file 2513px wide by 3263px high (8.375 by 10.875 inches; resolution of 600ppi). Please avoid graphs and small type.

If you have any questions or encounter any problems while uploading your accepted manuscript files, please email the editorial office at sourcefiles@thegsajournals.org.

Sincerely,

Sarah Otto
Associate Editor
GENETICS

Approved by:
David Begun
Senior Editor
GENETICS

note: Please add jnls.author.support@oup.com and genetics.oup@kwglobal.com (or the domains @oup.com and @kwglobal.com) to your email program's "safe senders" list. You will be contacted by both at various points during the production process.

Review comments (if applicable):